# SUMOylation of Bonus, the *Drosophila* homolog of Transcription Intermediary Factor 1, safeguards germline identity by recruiting repressive chromatin complexes to silence tissue-specific genes

**Baira Godneeva[1,2], Maria Ninova[3], Katalin Fejes-Toth[1], Alexei Aravin[1]\***

[1]California Institute of Technology, Division of Biology and Biological Engineering, Pasadena, United States; [2]Institute of Gene Biology, Russian Academy of Sciences, Moscow, Russian Federation; [3]University of California, Riverside, Riverside, United States

**\*For correspondence:**
aravin@caltech.edu

**Competing interest:** The authors declare that no competing interests exist.

**Abstract** The conserved family of Transcription Intermediary Factors (TIF1) proteins consists of key transcriptional regulators that control transcription of target genes by modulating chromatin state. Unlike mammals that have four TIF1 members, *Drosophila* only encodes one member of the family, Bonus. Bonus has been implicated in embryonic development and organogenesis and shown to regulate several signaling pathways, however, its targets and mechanism of action remained poorly understood. We found that knockdown of Bonus in early oogenesis results in severe defects in ovarian development and in ectopic expression of genes that are normally repressed in the germline, demonstrating its essential function in the ovary. Recruitment of Bonus to chromatin leads to silencing associated with accumulation of the repressive H3K9me3 mark. We show that Bonus associates with the histone methyltransferase SetDB1 and the chromatin remodeler NuRD and depletion of either component releases Bonus-induced repression. We further established that Bonus is SUMOylated at a single site at its N-terminus that is conserved among insects and this modification is indispensable for Bonus's repressive activity. SUMOylation influences Bonus's subnuclear localization, its association with chromatin and interaction with SetDB1. Finally, we showed that Bonus SUMOylation is mediated by the SUMO E3-ligase Su(var)2–10, revealing that although SUMOylation of TIF1 proteins is conserved between insects and mammals, both the mechanism and specific site of modification is different in the two taxa. Together, our work identified Bonus as a regulator of tissue-specific gene expression and revealed the importance of SUMOylation as a regulator of complex formation in the context of transcriptional repression.

## eLife assessment

This **important** study advances our knowledge of *Drosophila* Bonus, the sole ortholog of the mammalian transcriptional regulator Tif1. **Solid** evidence, both in vivo and in vitro, shows how SUMOylation controls the function of the Bonus protein and what the impact of SUMOylation on the function of Bonus protein in the ovary is.

## Introduction

Epigenetic regulation of gene expression is an essential mechanism that guides cell differentiation during development. The post-translational modifications of chromatin proteins act in combination with various chromatin-remodeling proteins to mediate changes in transcriptional activities and chromatin structure (reviewed in *Berger, 2007*; *Kouzarides, 2007*). TRIM/RBCC is an ancient protein family characterized by the presence of an N-terminal RING finger domain closely followed by one or two B-boxes and a coiled coil domain. Additional protein domains found at their C termini have been used to classify TRIM proteins into subfamilies. The Transcriptional Intermediary Factor 1 (TIF1) proteins present in Bilaterian species contain PHD and Bromo domains at their C-terminus and belong to Subfamily E according to *Marin, 2012* or structural class E according to *Ozato et al., 2008*. In vertebrates this subfamily contains four proteins: TIF1α/TRIM24, TIF1β/TRIM28, TIF1γ/TRIM33, and TIF1δ/TRIM66, while only one protein, Bonus (Bon), is present in *Drosophila*, making it an attractive model to understand the conserved functions of TIF1 proteins.

Mammalian TIF1 proteins are chromatin-associated factors that have been shown to play an essential role in transcription, cell differentiation, cell fate decisions, DNA repair, and mitosis (*Bai et al., 2010*; *Cammas et al., 2004*; *Cammas et al., 2000*; *Kulkarni et al., 2013*; *Le Douarin et al., 1996*; *Nielsen et al., 1999*; *Sedgwick et al., 2013*). TIF1 proteins modulate the transcription of target genes by binding to co-regulators in the genome and controlling the chromatin state (*Khetchoumian et al., 2004*; *Nielsen et al., 1999*; *Schultz et al., 2002*; *Schultz et al., 2001*; *Venturini et al., 1999*). One of the best characterized TIF1 proteins, KAP-1 (TIF1β), is the universal cofactor for the large family of Krüppel-associated box zinc-finger proteins (KRAB-ZFPs) composing one of the best-studied gene silencing systems in vertebrates (*Friedman et al., 1996*). Diverse KRAB-ZFPs recognize specific DNA sequences with the majority targeting endogenous retroviruses, ensuring their repression. After target recognition by KRAB-ZFPs, KAP-1 suppresses target transcription with the help of the H3K9-specific histone methyltransferase SetDB1, the H3K9me3 reader HP1, and the NuRD histone deacetylase complex (*Schultz et al., 2002*; *Schultz et al., 2001*).

The only member of the TIF1 subfamily in *Drosophila*, Bon was shown to be important in the development of several organs and somatic tissues during embryogenesis and metamorphosis, including the nervous system and the eye (*Allton et al., 2009*; *Beckstead et al., 2001*; *Ito et al., 2012*; *Kimura et al., 2005*; *Salzberg et al., 1997*; *Zhao et al., 2023*). Bon has been shown to regulate the function of different signaling pathways to drive developmental fate decisions, such as the ecdysone pathway (*Beckstead et al., 2001*) and the Hippo pathway in the eye (*Zhao et al., 2023*). Bon can act as both an Enhancer and a Suppressor of position-effect variegation (*Beckstead et al., 2005*), suggesting that it might play different roles that depend on specific interactors.

Many TRIM proteins from different subfamilies, including the mammalian TIF1γ/TRIM33, act as ubiquitin ligases, suggesting that this was the ancient function of the family. On the other hand, several members, including the mammalian KAP-1 was shown to be active as E3 SUMO-ligases. Furthermore, SUMOylation plays an essential role in KAP-1 function: KAP-1 is SUMOylated through its own activity and SUMOylation is required for its repressive function by facilitating recruitment of the SetDB1 histone methyltransferase (*Ivanov et al., 2007*; *Lee et al., 2007*; *Li et al., 2007*; *Mascle et al., 2007*). SUMO (small ubiquitin-like modifier) is a small protein that is covalently conjugated to lysine residues of substrates that can modify and enhance protein–protein interactions (*Gareau and Lima, 2010*; *Jentsch and Psakhye, 2013*; *Martin et al., 2007*). SUMOylation has been implicated in facilitating formation of protein complexes and condensates, especially in the nucleus, in different contexts including DNA repair, transcriptional repression and formation of subnuclear structures, and chromatin domains (reviewed in *Garvin and Morris, 2017*; *Gill, 2005*; *Verger et al., 2003*). The SUMO conjugation cascade involves the E1-activating enzyme, the E2-conjugating enzyme, and multiple E3-ligases that interact with E2 and facilitate the transfer of SUMO to the final substrates (*Gill, 2004*; *Johnson and Gupta, 2001*).

Here, we show that depletion of Bon in the female germline results in defective oogenesis and female infertility. We found that Bon controls oogenesis through repression of ectopic gene expression indicating that it serves as a guardian of cell-type identity. Mechanistically, we found that Bon induces transcriptional repression through interaction with the dNuRD chromatin remodeler and the SetDB1 histone methyltransferase. We show that Bon is SUMOylated at a single site at its N-terminus and that this modification is essential for Bon-induced transcriptional silencing. Furthermore,

this modification is important for Bon subnuclear localization and chromatin association as well as its interaction with SetDB1. The N-terminal SUMOylation site is conserved in insect species, but not in mammalian KAP-1 where several SUMOylation sites were reported at the C-terminal portion of the protein. Finally, we established that Bon SUMOylation depends on a distinct SUMO E3-ligase, Su(var)2–10, in contrast to mammalian KAP-1 that auto-SUMOylates itself. Our results identify Bon as a regulator of tissue-specific gene expression and highlight the universal function of SUMOylation as a regulator of complex formation in the context of transcriptional repression. On the other hand, our work suggests that SUMOylation of *Drosophila* Bon and mammalian KAP-1 has evolved independently and through distinct mechanisms revealing a remarkable case of parallel evolution in insects and vertebrates.

## Results

### *bon* knockdown in the female germline interferes with germline stem cells function and leads to arrested oogenesis and sterility

According to FlyAtlas, the *bon* gene encodes a nuclear protein that is expressed throughout development with high level of expression in several tissues including the brain, gut, and ovaries (FlyAtlas; *Chintapalli et al., 2007*). Immunostaining with antibodies against Bon revealed that it is expressed in both the germline and somatic cells at all stages of oogenesis, starting from the germarium which contains GSCs to late-stage egg chambers where GSC-derived nurse cells support maturing oocytes (*Figure 1A*). While *bon* was shown to be required for metamorphosis and the development of the nervous system (*Beckstead et al., 2001*; *Ito et al., 2012*), its function in the germline remained unknown. To gain insights into the germline functions of Bon, we generated transgenic flies expressing short hairpin RNAs (shRNAs) against *bon* under control of the UAS/Gal4 system and performed germline-specific RNAi knockdown (GLKD). Using the *maternal tubulin-Gal4 (MT-Gal4)* driver, we found by RT-qPCR (quantitative reverse transcription PCR) that two distinct shRNAs targeting *bon* led to 75% and 88% reduction in ovarian Bon expression, respectively (*Figure 1B*). Because the *MT-Gal4* driver is active in the germline, but not in follicular cells, the actual knockdown efficiency of *bon* in germ cells is even higher than what we detected from whole ovarian lysates. Indeed, immunofluorescence confirmed that Bon protein had been efficiently depleted from germline cells (*Figure 1C*). For all subsequent experiments, we used the shRNA that resulted in higher knockdown efficiency.

To analyze the role of Bon throughout the developmental progression of the germline we combined the *bon* shRNA construct with three different germline Gal4 drivers using different stage-specific promoters: *bam-Gal4*, which is expressed from cystoblasts to eight-cell cysts; *MT-Gal4*, which drives expression in germ cells starting in stage 2 of oogenesis, and *nos-Gal4* driver, which induces expression in two distinct stages, in GSCs and at late stages of oogenesis (*Figure 1—figure supplement 1A*; *Chen and McKearin, 2003*; *Van Doren et al., 1998*; *McKearin and Ohlstein, 1995*). Bon GLKD driven by either *MT-Gal4*, *bam-Gal4*, or the double driver (*MT + bam*) did not result in significant changes in ovarian morphology compared to controls (*Figure 1D*). Furthermore, such females laid eggs and were fertile. Thus, germline depletion of Bon starting at the cystoblast stage does not lead to morphological or obvious functional defects in oogenesis. In contrast, silencing of Bon beginning in the GSCs by expressing the shBon using *nos-Gal4* driver induces visible morphological changes with 34% of flies having only rudimentary ovaries lacking late stages of oogenesis and another ~39% having one of the two ovaries rudimentary (*Figure 1E*). An even stronger phenotype was observed upon GLKD using a double *nos + bam* driver which drives expression at all stages of oogenesis (*Figure 1D, F*). 100% of such females displayed rudimentary ovaries and were completely sterile (*Figure 1D, E*). Consistent with this, immunostaining for the germ cell marker Vasa demonstrates that depletion of Bon results in partial loss of germ cells and arrested oogenesis as morphological defects were accompanied by loss of vasa-positive cells from the egg chambers (*Figure 1F*). The loss of germ cells was further confirmed by the TUNEL assay which detects DNA fragmentation associated with cell death (*Figure 1—figure supplement 1B*). Additionally, we proved the importance of Bon in the *Drosophila* ovarian germline by using CRISPR/Cas9-mediated mutagenesis. Transgenic flies from the Heidelberg CRISPR Fly Design Library (*Port et al., 2020*) expressing sgRNAs targeting *bon*

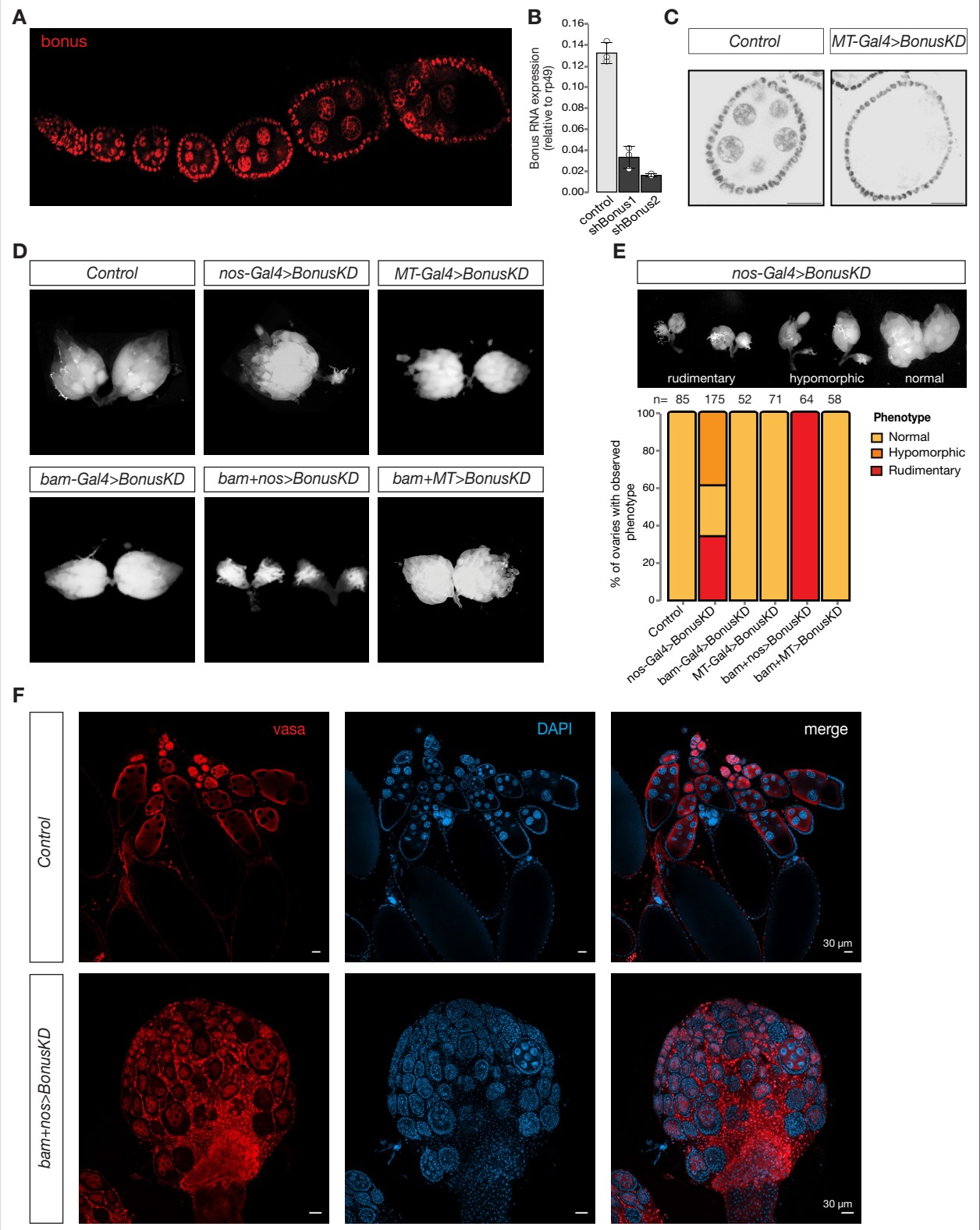

**Figure 1.** Germline expression of Bonus is required for oogenesis. (**A**) Bon is expressed throughout oogenesis. Stacked confocal image of wild-type Oregon-R flies stained for Bon. (**B**) Bar graph shows the relative expression of Bon (normalized to rp49 level) in control and Bon-depleted ovaries (RT-qPCR, dots correspond to three independent biological replicates (n=3); error bars indicate st. dev.; p<0.001, two-tailed Student's t-test). (**C**) Confocal images of egg chambers from wild-type Oregon-R flies (control) and flies expressing *MT-Gal4*-driven shRNA against Bon stained for Bon (scale bar:

*Figure 1 continued on next page*

*Figure 1 continued*

20 µm). (**D**) Bon depletion leads to rudimentary ovaries. Phase contrast images of dissected ovaries from flies of indicated genotypes. Wild-type Oregon-R flies were used as control. (**E**) Top: phase contrast image of dissected ovaries with different phenotypes from flies with Bon GLKD driven by *nos-Gal4*. Bottom: graph showing the percentage of normal, hypomorphic, and rudimentary ovary phenotypes of indicated genotypes (*n* = 85, 175, 52, 71, 64, and 58, respectively). (**F**) Confocal images of whole ovaries from wild-type Oregon-R flies (control) and flies with Bon GLKD driven by *bam + nos* double driver stained for Vasa (red) and DAPI (4′,6-diamidino-2-phenylindole) (blue) (scale bar: 30 µm).

The online version of this article includes the following figure supplement(s) for figure 1:

**Figure supplement 1.** An important function of Bonus in the early stages of oogenesis.

were crossed to *nos-Gal4;UAS-Cas9* to achieve germline-specific knockout of *bon*. Almost 65% of the female offspring with *nos-Cas9;sgRNA-bon* were sterile and had defects in ovarian morphology, another 23% had one rudimentary ovary and only 12% showed normal phenotype (*Figure 1—figure supplement 1C*). These results indicate that Bon function in the early stages of oogenesis, particularly in GSCs, is essential for proper oogenesis.

To further analyze the role of Bon in the maintenance of GCSs and early oogenesis we used immunofluorescence against the cytoskeletal protein α-spectrin, which marks the spectrosome, a spherical intracellular organelle present in GSCs and cystoblasts. At later stages, spectrosomes become fusomes, branched structures that are localized in cytoplasmic bridges connecting differentiating germ cells in the growing cysts. Thus, fusome formation is a hallmark of normal oogenesis progression. In ovarioles of control flies, we observed a normal germarium organization with two to three spectrosome-containing GSCs, and branched fusomes in germ cells at later stages. In contrast, germ cells with normal fusomes were absent upon depletion of Bon using *nos-Gal4*. Instead, the germarium of Bon-depleted flies harbored several cells containing spherical spectrosomes, a hallmark of GSCs or cystoblast-like undifferentiated germ cells (*Figure 1—figure supplement 1D, E*). Overall, our results indicate that loss of Bon in early germ cells interferes with maintenance of GSCs and arrests their further differentiation.

## Loss of Bonus triggers the ectopic expression of tissue-specific genes in the ovary

To investigate the effect of Bon depletion on gene expression in the female germline, we performed transcriptome profiling using RNA sequencing (RNA-seq) analysis. We tested the effects of loss of Bon in both early and later stages of oogenesis using the *nos-Gal4* or *MT-Gal4* driver to drive *bon* shRNA expression, respectively. RNA-seq libraries were prepared in triplicates and compared to respective control libraries. As knockdown using the *nos-Gal4* driver causes early arrest of oogenesis and rudimentary ovaries, while later-stage knockdown with the *MT-gal4* driver yields normal ovaries, we used different controls depending on the driver to assure that ovary size and cell composition of the Bon GLKD and control are similar. For *nos-Gal4* we used ovaries from young (0- to 1-day old) flies that lack later stages of oogenesis and compared them to their age-matched siblings that lack the shRNA, and for *MT-gal4* we used 1- to 2-old flies that express either shRNA against *bon* or the *white* gene, which is not expressed in the germline. Thus, in both cases, GLKD and control flies had the same age and similar ovary size. As the mammalian homolog of Bon, KAP-1, plays a central role in repression of many transposable elements (TEs) through its function as co-repressor for multiple KRAB-ZFPs that recognize TEs sequences, we analyzed expression of both host genes and TEs.

Most TEs families were not affected by Bon depletion using either driver. Using the *nos-Gal4*-driven shRNA, only 6 out of 207 (~3%) TE families present in the *Drosophila* genome significantly increased their expression more than twofold (log$_2$FC >1, and qval <0.05, LRT test (Likelihood Ratio Test), Sleuth) (*Figure 2—figure supplement 1A*) and none showed strong (>sixfold) upregulation. Similarly, depletion of Bon at later stages of oogenesis also did not lead to strong (>sixfold) change in transposon expression (*Figure 2—figure supplement 1A*). This phenotype is in stark contrast to the significant activation of many TE families when the main TE repression pathway in the ovary – the piRNA pathway – is abolished, suggesting that Bon is not involved in the piRNA pathway and that oogenesis defects observed upon Bon depletion likely have a different molecular basis.

In contrast to TEs, the protein-coding transcriptome was severely disrupted upon Bon depletion – differential gene expression analysis using Sleuth revealed many genes with altered steady-state RNA levels upon Bon GLKD at either stage. As expected, Bon was one of the most strongly downregulated

genes, showing ~sevenfold reduction and confirming the efficiency of KD and the validity of the RNA-seq data (*Figure 2A*). Early Bon GLKD resulted in 694 differentially expressed genes (qval <0.05, LRT test) (*Figure 2A*), of which 464 (~67%) and 28 (~4%) genes, respectively, increased and decreased their expression more than twofold, while late Bon GLKD using *MT-Gal4* revealed 1769 genes that were differentially expressed (qval <0.05, LRT test), with 231 genes (~13.6%) showing more than twofold increase, while 72 genes (~4.2%) showed more than a twofold decrease in mRNA level (*Figure 2—figure supplement 1B*). Interestingly, the sets of genes that change their expression upon Bon GLKD at the early and late stages of oogenesis are quite different: only 51 genes were derepressed at both stages of oogenesis, while the remaining genes that changed their expression were unique for one or the other stage (*Figure 2—figure supplement 1C*). Overall, our results indicate that Bon plays an important role in regulation of gene expression during oogenesis with distinct targets at different stages.

To characterize Bon targets in oogenesis, we performed gene ontology (GO) analysis of genes strongly upregulated upon *nos-Gal4*-driven Bon GLKD (*n* = 464). GO analysis identified enrichment of genes from 27 biological process in the set of Bon-repressed genes (*Figure 2B*). These included terms such as mesoderm development, myofibril assembly, sarcomere organization, hemolymph coagulation, motor neuron axon guidance, and visceral muscle development, suggesting that Bon loss leads to the ectopic ovarian activation of genes normally expressed in other tissues. To comprehensively explore the specific expression patterns of the 464 Bon-repressed genes, we used modENCODE RNA-seq data from different tissues. This analysis revealed that many genes that are derepressed in the ovary upon Bon GLKD are normally expressed in other tissues and have no (55%) or low (33%) expression in the ovary of wild-type flies. Instead, many of these genes are predominantly expressed in the head (50%), digestive system (38%), and central nervous system (28%) of wild-type flies (*Figure 2C*).

We used RT-qPCR and in situ hybridization chain reaction (HCR) for selected upregulated genes including *rbp6*, *CG34353*, and *ple*, which are highly expressed in the head, and *pst*, which is highly expressed in the gut, to confirm that germline depletion of Bon triggers their ectopic activation. No signal for these genes was detected in wild-type ovaries, while abundant *rbp6*, *CG34353*, and *pst* transcripts were identified in germ cells upon *MT-Gal4>Bon* GLKD (*Figure 2D-F*, *Figure 2—figure supplement 1D-F*). Surprisingly, depletion of Bon in germ cells caused the appearance of *ple* transcripts in somatic follicular cells that surround germline cells, suggesting that Bon depletion causes activation of *ple* indirectly, through a process that involves signaling between the adjacent germline and follicular cells (*Figure 2D*). Overall, our results indicate that in the ovary, Bon is required for repression of genes that are typically expressed in non-ovarian tissues.

## Recruitment of Bonus to a genomic locus induces transcriptional repression associated with accumulation of the H3K9me3 mark

Transcriptome profiling upon Bon germline depletion demonstrated global changes in steady-state RNA levels of hundreds of genes. As exemplified by the activation of the *ple* gene in the somatic follicular cells some of these effects might be indirect and even mediated by intercellular signaling. To test the ability of Bon to directly induce transcriptional silencing, we took advantage of a tethering approach in which Bon is recruited to a reporter locus via binding to nascent transcripts (*Figure 3A*). Tethering was achieved through fusion of Bon to the $\lambda$N RNA-binding domain that has high affinity for BoxB RNA hairpins encoded in the 3'UTR region of the reporter gene (*De Gregorio et al., 1999*). $\lambda$N-eGFP-Bon and the reporter were co-expressed in the germline using the *MT-Gal4* driver; recruitment of $\lambda$N-eGFP was used as a control.

RT-qPCR showed that tethering of Bon triggers ~22-fold reporter repression (*Figure 3B*). Similar results were obtained with a different reporter in another genomic location, indicating that recruitment of Bon induces strong repression regardless of the genomic locus (*Figure 3—figure supplement 1A*). ChIP-qPCR analysis revealed that Bon recruitment results in a strong increase in the repressive H3K9 trimethylation (H3K9me3) chromatin mark, at the reporter locus (*Figure 3C*), suggesting that repression induced by Bon is mediated, at least in part, by the deposition of H3K9me3.

We also examined changes in H3K9me3 enrichment on genes upregulated upon Bon depletion (Bon GLKD driven by *nos-Gal4*). Global ChIP-seq analysis revealed that many Bon-dependent genes show low or no H3K9me3 signal in control ovaries and no change upon Bon depletion, hence might be secondary targets (*Figure 3D*, *Figure 3—figure supplement 1B*). For instance, the gene *pst* despite

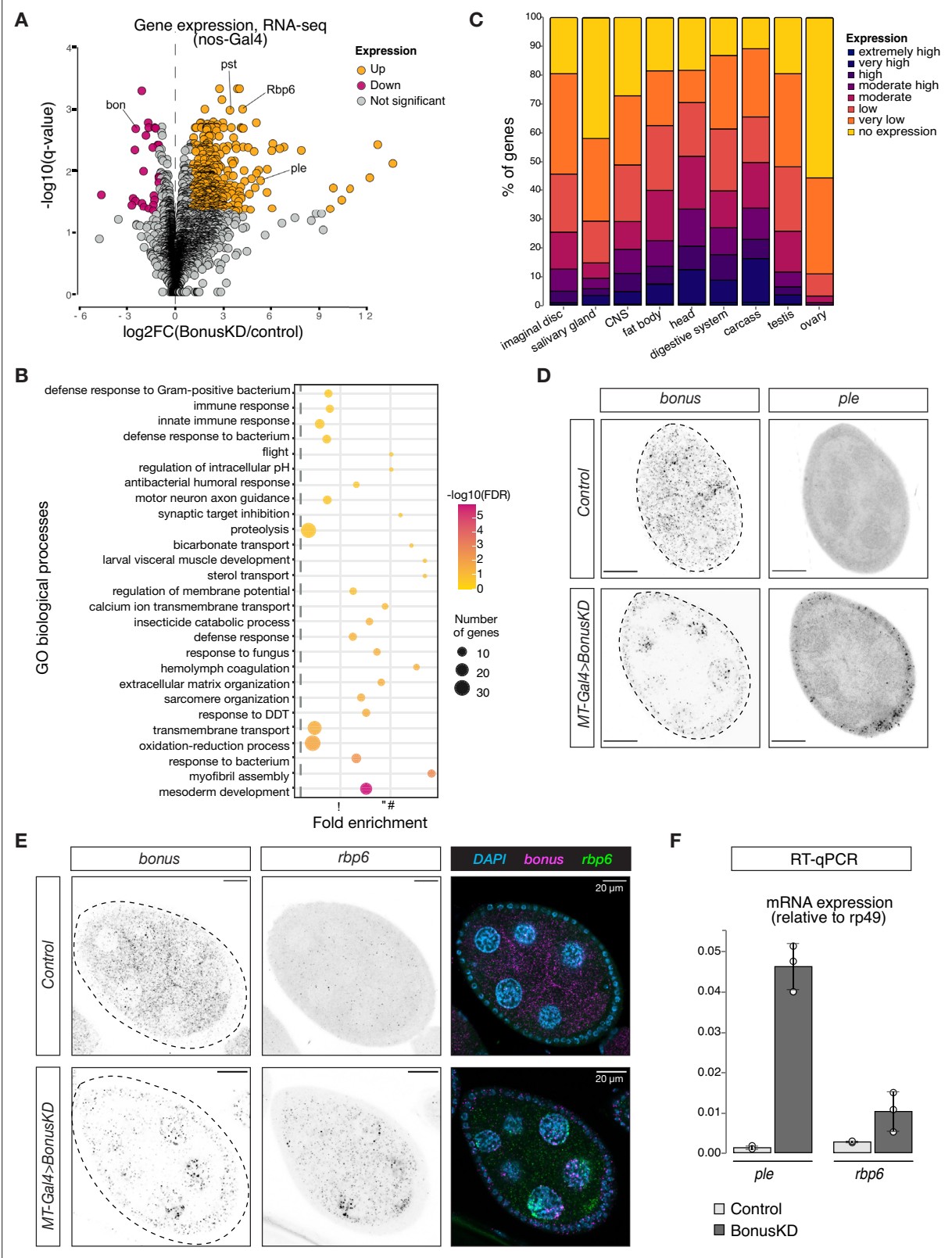

**Figure 2.** Bonus functions as a repressor of tissue-specific genes in ovary. (**A**) Bon GLKD leads to misexpression of tissue-specific genes in the ovary. Volcano plot shows fold changes in genes expression upon Bon GLKD driven by *nos-Gal4* in the ovary as determined by RNA-seq (*n* = 3). Siblings that lack shRNA against Bon produced in the same cross were used as a control. Genes that change significantly (log$_2$FC >1, qval <0.05, LRT test, sleuth; **Pimentel et al., 2017**) are highlighted. Genes *bon*, *pst*, *Rbp6*, and *ple* are labeled. Genes with infinite fold change values (zero counts in control

*Figure 2 continued on next page*

*Figure 2 continued*

ovaries) are not shown. (**B**) Bon represses genes with diverse functions. Bubble plot shows the analysis of gene ontology (GO) enrichment at the level of biological processes (BP) for genes that are derepressed upon Bon GLKD driven by *nos-Gal4* (log$_2$FC >1, qval <0.05, LRT test, sleuth; *Pimentel et al., 2017*). Only GO terms above the established cut-off criteria (p-value <0.01 and >3 genes per group) are shown. BP are ranked by fold enrichment values. The most significant processes are highlighted in purple, and the less significant in yellow according to log$_{10}$(FDR) values. The bubbles size reflects the number of genes, assigned to the GO BP terms. (**C**) Normal expression level of deregulated genes upon Bon GLKD in the tissues where they are normally expressed indicates Bon-mediated silencing of genes normally expressed in the head and digestive system. The graph shows the percentage of derepressed genes upon Bon GLKD driven by *nos-Gal4* (log$_2$FC >1, qval <0.05, LRT test, sleuth; *Pimentel et al., 2017*) with given expression level in the indicated enriched tissues. Expression levels according RPKM values from modENCODE anatomy RNA-seq dataset are no expression (0–0), very low (1–3), low (4–10), moderate (11–25), moderate high (26–50), high (51–100), very high (101–1000), and extremely high (>1000). (**D**) GLKD of Bon leads to *ple* expression in follicular cells. Confocal images of egg chambers show RNA in situ hybridization chain reaction (HCR) detecting *ple* and *bonus* mRNAs in flies with *MT-Gal4>Bon* GLKD and control siblings from the same cross that lack Bon shRNA (scale bar: 20 µm). (**E**) Bon represses *rbp6* in the germline. Confocal images of egg chambers show RNA in situ HCR detecting *rbp6* and *bonus* mRNAs in flies with *MT-Gal4>Bon* GLKD and control siblings from the same cross that lack Bon shRNA (scale bar: 20 µm). (**F**) Bar graph shows the relative expression of *ple* and *rbp6* (normalized to rp49 level) in control and Bon-depleted ovaries (RT-qPCR, dots correspond to three independent biological replicates (n=3); error bars indicate st. dev.).

The online version of this article includes the following figure supplement(s) for figure 2:

**Figure supplement 1.** Depletion of Bon induces ectopic activation of non-ovarian genes.

being activated upon Bon GLKD displayed a low H3K9me3 signal (*Figure 3—figure supplement 1B*). However, several Bon-regulated genes are enriched in H3K9me3 mark in wild-type ovaries including in the proximity of the transcription start site (TSS) and show prominent loss of H3K9me3 upon Bon depletion (*Figure 3D*). For example, gene *CG1572*, which was activated twofold upon Bon GLKD, showed almost a twofold decrease in H3K9me3 level upstream of its TSS (*Figure 3E*). Independent ChIP-qPCR analysis of few Bon-regulated genes such as *CG3191* and *Spn88Eb* also showed a slight decrease in the repressive mark upon Bon depletion (*Figure 3F*).

Altogether, these data indicate that Bon recruitment to genomic targets induces transcriptional repression associated with accumulation of the H3K9 trimethylation mark. However, repression of many Bon-regulated genes might be indirect and/or independent of H3K9me3.

## Bonus interacts with dNuRD complex components Mi-2 and Rpd3, as well as the histone methyltransferase SetDB1

Mammalian KAP-1 was shown to associate with the NuRD histone deacetylase and chromatin-remodeling complex and with the H3K9me3 writer SetDB1, and their interactions are important for its function in transcriptional repression (*Schultz et al., 2002*; *Schultz et al., 2001*). In *Drosophila*, the dNuRD complex mediates chromatin remodeling and histone deacetylation through dMi-2 and Rpd3 (HDAC1 homolog), respectively (*Bouazoune and Brehm, 2006*; *Brehm et al., 2000*; *Kunert and Brehm, 2009*; *De Rubertis et al., 1996*; *Tong et al., 1998*). To study whether dNuRD and SetDB1 are required for Bon's ability to trigger transcriptional repression in the *Drosophila* germline, we tested reporter expression upon Bon tethering and concomitant knockdown of SetDB1 and dNuRD components. GLKD of either Mi-2 or SetDB1, but not Rpd3, inhibited silencing, indicating that Mi-2 and SetDB1 act downstream of Bon to induce repression (*Figure 4A*). Notably, we found that 29% of the derepressed genes (135 out of the 464 genes) overlap with those upregulated in *nos-Gal4*-driven SetDB1 GLKD, suggesting that Bon and SetDB1 co-regulate many genes.

To explore physical interactions of Bon with components of the dNuRD and SetDB1 complexes, we employed co-immunoprecipitation assay using tagged proteins in S2 cells. We found that both components of dNuRD, Mi-2, and Rpd3, as well as SetDB1 co-purify with Bon (*Figure 4B-D*, *Figure 4—figure supplement 1A*). The interaction between Bon and Mi-2 is mediated by the C-terminus of Mi-2 (*Figure 4D*), similar to interaction between Mi-2α/CHD3 and KAP-1 in mammals (*Schultz et al., 2001*). In *Drosophila* Mi-2 is found in two distinct complexes, the canonical dNuRD complex and the dMec complex that contains the zinc-finger protein Mep-1 (*Kunert et al., 2009*). We did not detect an interaction between Bon and Mep-1 (*Figure 4—figure supplement 1B*), indicating that Bon interacts with Mi-2 in the context of the dNuRD complex but not dMec. Overall, our results indicate that the interactions between Bon and the NuRD and SetDB1 chromatin remodeler and modifying complexes are

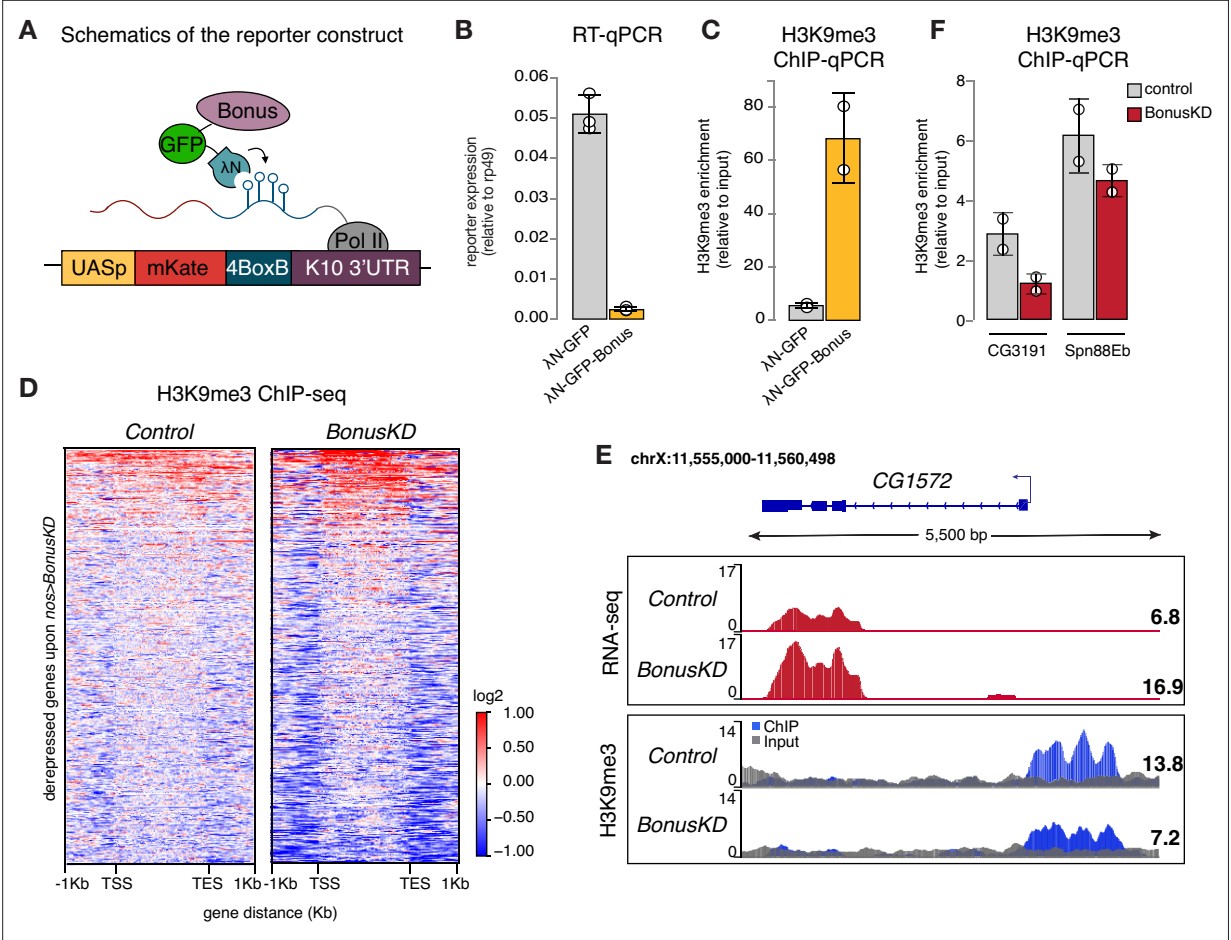

**Figure 3.** Bonus induces transcriptional silencing. (**A**) Schematics of the reporter construct in flies that allows Bon recruitment to nascent reporter transcript in flies. λ N-GFP-Bonus and the mKate reporter encoding 4BoxB hairpins are co-expressed in germline cells of the ovary (driven by *MT-Gal4*). (**B**) Bon tethering leads to transcriptional silencing of the reporter. Bar plot shows reporter expression (normalized to rp49 level) upon tethering of λ N-GFP-Bonus or λ N-GFP control ovaries (RT-qPCR, dots correspond to three independent biological replicates (n=3); error bars indicate st. dev.; p<0.001, two-tailed Student's t-test). (**C**) Bon tethering leads to H3K9me3 accumulation. Bar plot shows H3K9me3 enrichment upon tethering of λ N-GFP-Bonus or λ N-GFP control ovaries (ChIP-qPCR, dots correspond to two independent biological replicates (n=2); error bars indicate st. dev.; p<0.05, two-tailed Student's t-test). (**D**) Heatmap shows H3K9me3 distribution across Bon targets in control and *nos-Gal4>Bon* GLKD ovaries (input-normalized $\log_2$ values). (**E**) RNA-seq and ChIP-seq tracks show counts per million (CPM)-normalized coverage for *CG1572* in control and *nos-Gal4>Bon* GLKD ovaries. The gene structure is depicted at the top; arrow indicates the direction of transcription. The ChIP (blue) and input (gray) signals are overlaid. Numbers show the CPM values of the exonic regions (RNA-seq) or the normalized ChIP/input signal (ChIP-seq) in a manually selected genomic location. (**F**) Bon depletion results in a slight decrease in H3K9me3 over some Bon target genes. Bar graph shows H3K9me3 levels at the genes *CG3191* and *Spn88Eb* in control and Bon-depleted ovaries (ChIP-qPCR, dots correspond to two independent biological replicates (n=2); error bars indicate st. dev.).

The online version of this article includes the following figure supplement(s) for figure 3:

**Figure supplement 1.** Bonus tethering leads to transcriptional silencing of the reporter.

essential for its repressor activity and evolutionarily conserved among members of the TIF1 protein family between insects and mammals.

## Bonus is SUMOylated at a single site close to its N-terminus

Self-SUMOylation of mammalian KAP-1 is essential for its repressive function (*Ivanov et al., 2007*). In our analysis of immunopurified Bon by Western blotting, we noticed a band of higher molecular weight, indicative of a post-translationally modified form. To explore if Bon is SUMOylated, we co-expressed tagged SUMO and Bon in S2 cells followed by immunoprecipitation of Bon under stringent washing conditions to remove non-covalently bound proteins in the presence of *N*-ethylmaleimide (NEM), an inhibitor of SUMO-specific deconjugating enzymes. Western blot revealed the presence

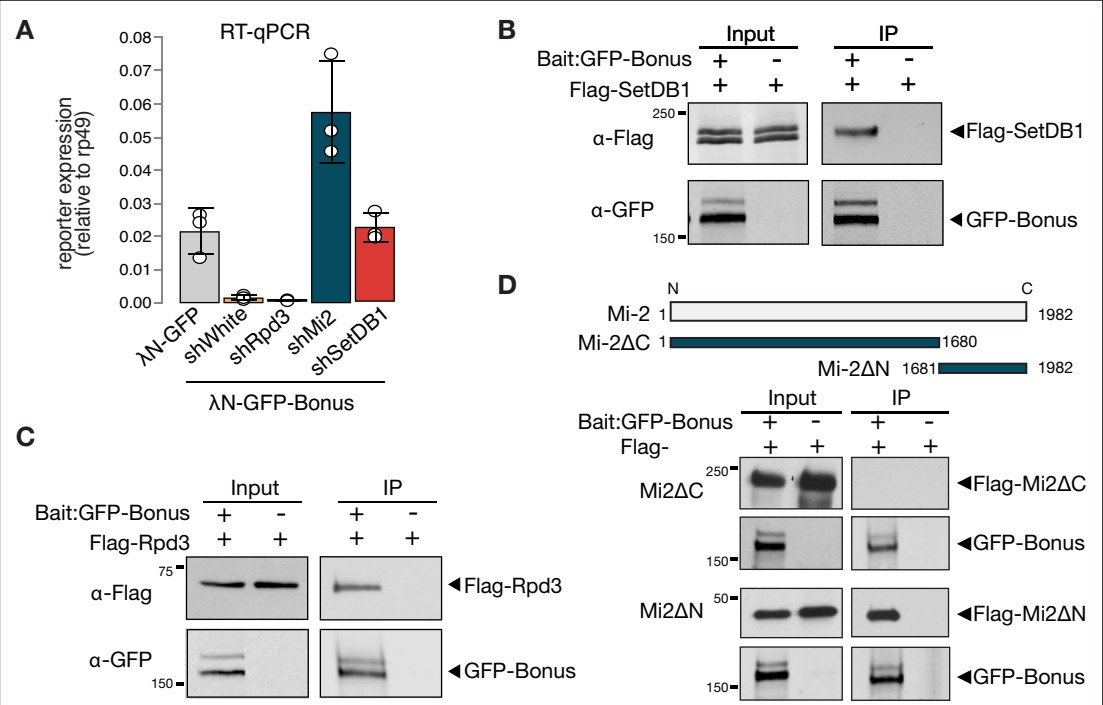

**Figure 4.** Bonus interacts with Mi-2, Rpd3, and SetDB1. (**A**) Reporter silencing by Bon depends on Mi-2 and SetDB1. Bar plot showing the reporter expression (normalized to rp49 level) upon tethering of control λN-GFP or λN-GFP-Bonus in ovaries with Rpd3, Mi-2, SetDB1 GLKD, and control *white* GLKD (RT-qPCR, dots correspond to three independent biological replicates (n=3); error bars indicate st. dev.). Bon interacts with SetDB1 and Rpd3. Western blot analysis of immunoprecipitation experiment using GFP nanotrap beads from S2 cells co-expressing GFP-Bonus and Flag-tagged SetDB1 (**B**) and Flag-tagged Rpd3 (**C**). Lysates not expressing GFP-Bonus were used as negative control. (**D**) Bon interacts with the C-terminus of Mi-2. Top: schematic illustration of full-length *Drosophila* Mi-2 and its truncated versions as defined by the amino acids: C-terminal truncated Mi-2 (1–1680) and N-terminal truncated Mi-2 (1681–1982). Bottom: western blot analysis of immunoprecipitation experiment using GFP nanotrap beads from S2 cells co-expressing GFP-Bonus and Flag-tagged Mi2 fragments. Lysate not expressing GFP-Bonus was used as negative control.

The online version of this article includes the following source data and figure supplement(s) for figure 4:

**Source data 1.** Annotated and uncropped western blots and raw images for *Figure 4B*.

**Source data 2.** Annotated and uncropped western blots and raw images for *Figure 4C*.

**Source data 3.** Annotated and uncropped western blots and raw images for *Figure 4D*.

**Figure supplement 1.** Bonus interacts with Mi-2.

**Figure supplement 1—source data 1.** Annotated and uncropped western blots and raw images for *Figure 4—figure supplement 1A*.

**Figure supplement 1—source data 2.** Annotated and uncropped western blots and raw images for *Figure 4—figure supplement 1B*.

of unmodified and single SUMO-modified forms of Bon (*Figure 5—figure supplement 1A*, top). To explore if Bon is SUMOylated in fly ovaries, we immunoprecipitated Bon from ovarian extracts of flies that express Flag-tagged SUMO in the germline. As in S2 cells, we observed unmodified and single SUMO-modified Bon forms, indicating that a fraction of the Bon protein pool is SUMOylated in both S2 cells and ovaries (*Figure 5—figure supplement 1A*, bottom). In addition, we explored if Bon undergoes ubiquitination. Immunoprecipitation of Bon from ovarian extracts, followed by western blot using an anti-ubiquitin antibody did not reveal the presence of ubiquitinated form of Bon (*Figure 5—figure supplement 1C*).

To find potential SUMOylation sites in Bon we used the SUMOplot Analysis Program which yielded three high-scoring predicted residues: two sites, lysine K9 (L<u>K</u>ND) and K20 (I<u>K</u>QE), are conforming to the canonical consensus site for SUMOylation, ΨKxD/E, whereas the third one, K763, resides in a noncanonical motif, L<u>K</u>SP (*Figure 5A*). Unlike wild-type Bon, the triple mutant with all three lysine residues substituted to arginine was not SUMOylated when expressed in either S2 cells or fly ovaries (*Figure 5B*, *Figure 5—figure supplement 1B*). To further narrow the modification site, we created individual point mutants and checked their SUMOylation. The single K20R mutation completely

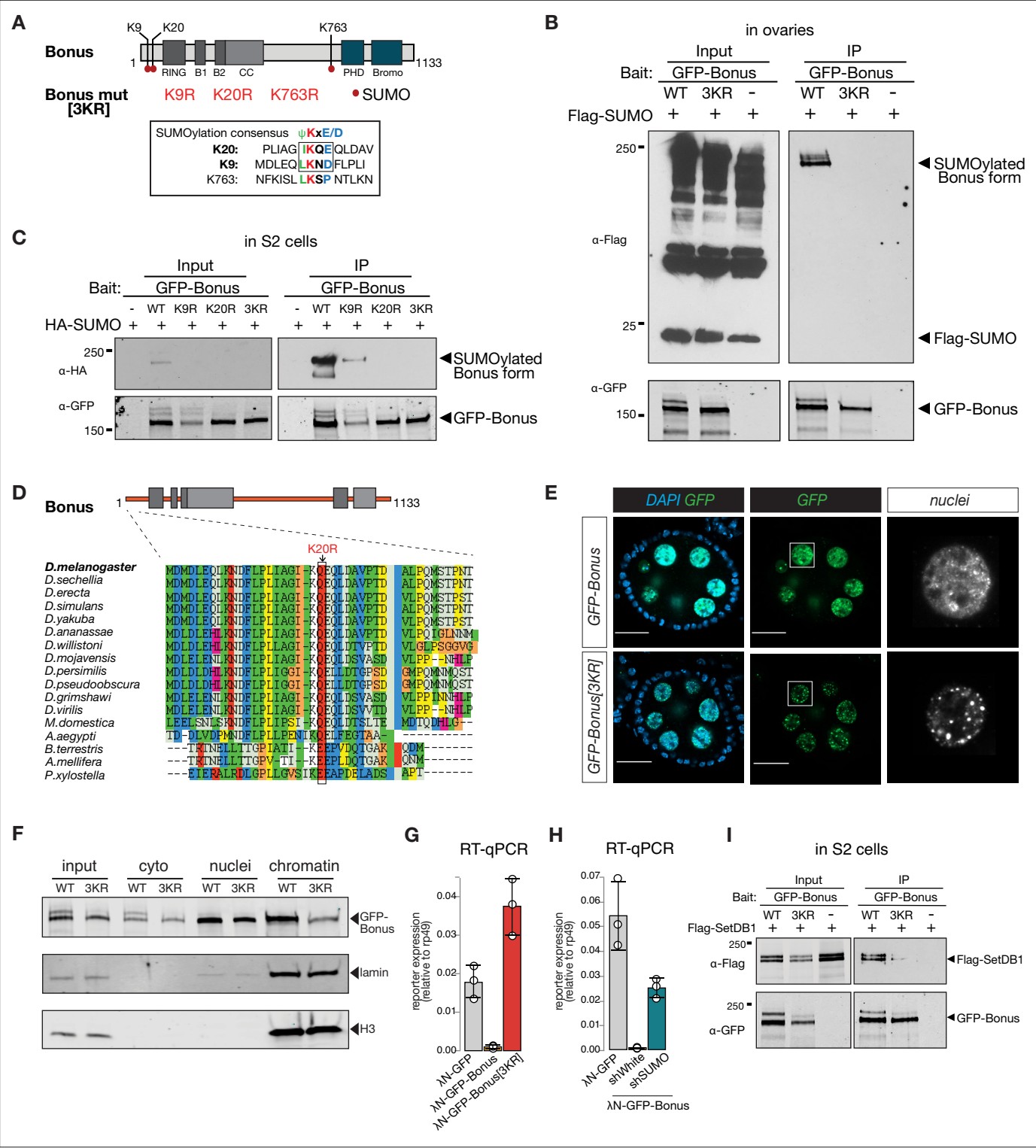

**Figure 5.** Bonus is SUMOylated. (**A**) Schematic representation of putative SUMOylation sites within Bon. SUMOylation consensus sites are shown and boxed. Canonical consensus sites are in bold. Putative SUMOylated lysines were mutated to arginines individually (K9R, K20R, and K763R) or in combination (3KR). (**B**) Bon is SUMOylated at specific residues. Western blot analysis shows the SUMOylation levels of GFP-tagged Bon and SUMO-deficient triple mutant 3KR expressed in fly ovaries. SUMOylated form of Bon was detected only in wild-type GFP-Bonus (WT). Total protein lysates from flies co-expressing Flag-SUMO and λN-GFP-Bonus or λN-GFP-Bonus[3KR] were immunopurified using anti-GFP nanotrap beads. Flies not expressing λN-GFP-tagged protein were used as a negative control. (**C**) Bon is predominantly SUMOylated at K20. Western blot analysis shows the associated SUMOylation levels of GFP-tagged Bon and SUMO-deficient triple mutant 3KR and single mutated K9R, K20R expressed in S2 cells. Single

*Figure 5 continued on next page*

*Figure 5 continued*

mutation K9R reduced, while the K20R mutation and triple 3KR mutation completely abolished Bon SUMOylation. Total protein lysates from S2 cells co-expressing HA-SUMO and GFP-Bonus or GFP-Bonus[3KR], GFP-Bonus[K9R], GFP-Bonus[K20R] were immunopurified using anti-GFP nanotrap beads. Lysate not expressing GFP-tagged protein was used as a negative control. (**D**) SUMOylation site of Bon is conserved in insects. Sequence alignment of the Bon protein sequence from 12 *Drosophila* species and other insects shows conserve action of canonical SUMOylation consensus at K20 (boxed and indicated by the arrowhead). (**E**) SUMO-deficient Bon mislocalizes into nuclear foci. Confocal images of egg chambers show the localization of *MT-Gal4*-driven λ N-GFP-tagged Bonus and SUMO-deficient triple mutant λ N-GFP-Bonus[3KR] flies. Images on the right panel show isolated nurse cell nuclei (scale bar: 20 μm). (**F**) Chromatin association of Bon depends on its SUMOylation. Western blot analysis shows the fractionation of cytoplasmic (cyto), nuclear (nuclei), and chromatin compartments of *MT-Gal4*-driven λ N-GFP-tagged Bonus (WT) and SUMO-deficient triple mutant λ N-GFP-Bonus (3KR) fly ovaries. Lamin and Histone H3 were used as markers for nuclear and chromatin fractions. (**G**) Bon-mediated reporter repression depends on Bon SUMOylation. Bar plot shows the reporter expression (normalized to rp49 level) upon tethering of λ N-GFP-Bonus, SUMO-deficient triple mutant λ N-GFP-Bonus[3KR] or λ N-GFP control ovaries (RT-qPCR, dots correspond to three independent biological replicates (n=3); error bars indicate st. dev.). (**H**) Bar plot shows the reporter expression (normalized to rp49 level) upon tethering of control λ N-GFP or λ N-GFP-Bonus in ovaries with SUMO GLKD, and control *white* GLKD (RT-qPCR, dots correspond to three independent biological replicates (n=3); error bars indicate st. dev.). (**I**) Western blot analysis shows the SUMO-dependent interaction between Bon and SetDB1. Total protein lysates from S2 cells co-expressing Flag-SetDB1 and GFP-Bonus (WT) or triple mutant GFP-Bonus[3KR] (3KR) were immunopurified using anti-GFP nanotrap beads. Lysate from cells not expressing GFP-tagged protein was used as a negative control.

The online version of this article includes the following source data and figure supplement(s) for figure 5:

**Source data 1.** Annotated and uncropped western blots and raw images for *Figure 5B*.

**Source data 2.** Annotated and uncropped western blots and raw images for *Figure 5C*.

**Source data 3.** Annotated and uncropped western blots and raw images for *Figure 5F*.

**Source data 4.** Annotated and uncropped western blots and raw images for *Figure 5I*.

**Figure supplement 1.** SUMOylation of Bonus.

**Figure supplement 1—source data 1.** Annotated and uncropped western blots and raw images for *Figure 5—figure supplement 1A*.

**Figure supplement 1—source data 2.** Annotated and uncropped western blots and raw images for *Figure 5—figure supplement 1B*.

**Figure supplement 1—source data 3.** Annotated and uncropped western blots and raw images for *Figure 5—figure supplement 1C*.

**Figure supplement 1—source data 4.** Annotated and uncropped western blots and raw images for *Figure 5—figure supplement 1D*.

**Figure supplement 2.** SUMO-independent and dependent interactions of Bonus.

**Figure supplement 2—source data 1.** Annotated and uncropped western blots and raw images for *Figure 5—figure supplement 2A*.

**Figure supplement 2—source data 2.** Annotated and uncropped western blots and raw images for *Figure 5—figure supplement 2B*.

**Figure supplement 2—source data 3.** Annotated and uncropped western blots and raw images for *Figure 5—figure supplement 2C*.

abolished Bon SUMOylation (*Figure 5C*). Moreover, examination of Bon SUMOylation in cell extracts using the SUMO protease SENP2 consistently revealed that Bon is SUMOylated at the single site (*Figure 5—figure supplement 1D*). Together with the characteristic shift in modified Bon migration on sodium dodecyl sulfate–polyacrylamide gel (SDS–PAGE), these results indicate that the bulk of SUMO-modified Bon carries a single SUMO moiety at the K20 residue. To explore whether this SUMOylation site is conserved in other *Drosophila* species, we performed an alignment of predicted Bon homologs from the genomes of 12 fully sequenced species using ClustalW. We found that the consensus SUMOylation site at the K20 position of *D. melanogaster* Bon is a conserved in all analyzed *Drosophila* species (*Figure 5D*, *Figure 5—figure supplement 1E*). Furthermore, analysis revealed conservation of the position of SUMOylated lysine residue in distantly related insects including the buff-tailed bumblebee *Bombus terrestris*, the honeybee *Apis mellifera*, yellow fever mosquito *Aedes aegypti*, and diamondback moth *Plutella xylostella* (*Figure 5D*), suggesting that SUMOylation at the Bon N-terminus is conserved across many insects. On larger evolutionary distances, although SUMOylation is conserved between the mammalian KAP-1 and the *Drosophila* Bon, its position is different: while mammalian KAP-1 is SUMOylated at multiple sites that all reside in its C-terminus, Bon is SUMOylated at a single site close to its N-terminus.

We compared the subcellular localization of wild-type and SUMO-deficient Bon in the germline. Wild-type Bon is localized in nurse cell nuclei and overall nuclear localization was not affected by lack of SUMOylation (*Figure 5E*). Wild-type Bon shows enrichment in some nuclear regions, but is generally distributed throughout the nucleus. ~37% of Bon colocalized with DAPI-dense chromatin regions (*Figure 5—figure supplement 1F*), however, Bon showed poor (8%) overlap with HP1 protein, the mark of gene-poor and repeat-reach heterochromatin (*Nakayama et al., 2001*; *Rea et al., 2000*),

indicating that Bon is not localized to the bulk of heterochromatin (*Figure 5—figure supplement 1F*, bottom). Remarkably, the lack of SUMOylation affected Bon distribution within the nucleus: mutant Bon was localized in more discrete nuclear foci compared to the wild-type protein (*Figure 5E*). To explore whether SUMOylation affects Bon localization on chromatin we separated the chromatin fraction and probed Bon presence by Western blot. SUMO-deficient mutant exhibited significantly reduced association with the chromatin fraction (*Figure 5F*), indicating that SUMOylation contributes to Bon's subnuclear localization and chromatin association. Both the unSUMOylated and SUMOylated forms of Bon were detected in the cytoplasmic fraction, suggesting that a small fraction of SUMOylated Bon may be exported from the nucleus to the cytoplasm.

To test the functional role of Bon SUMOylation, we explored the ability of SUMO-deficient Bon to induce transcriptional repression in the tethering assay. Unlike wild-type Bon, tethering of SUMO-deficient protein did not trigger repression of the reporter (*Figure 5G*). Furthermore, knockdown of *smt3*, the single gene encoding SUMO in the *Drosophila* genome, resulted in a partial release of the reporter silencing caused by tethering of wild-type Bon (*Figure 5H*). Thus, SUMOylation seems to be essential for the ability of Bon to induce transcriptional repression. Next, we tested if SUMOylation affects Bon interaction with SetDB1 and dNuRD. Immunoprecipitation assays showed that SUMOylation is dispensable for Bon interaction with Mi-2 and Rpd3 (*Figure 5—figure supplement 2A, B*, B), however, it decreased its interaction with SetDB1 (*Figure 5I*). Furthermore, mass spectrometry analysis of Bon-bound proteins in ovary revealed enrichment of SetDB1 in association with wild-type, but not SUMO-deficient Bon (enrichment level = 2.5). However, immunoprecipitation of SetDB1 showed that it primarily interacts with unmodified Bon which is much more abundant compared to SUMOylated protein (*Figure 5—figure supplement 2C*). Overall, our results revealed that SUMOylation of Bon at a single site modulates its subnuclear localization and is important for Bon's ability to induce transcriptional silencing and interact with SetDB1.

## Bonus SUMOylation depends on the SUMO E3-ligase Su(var)2–10

To test if Bon acts as a SUMO E3-ligase and can promote self-SUMOylation similar to mammalian KAP-1, we explored its interaction with Ubc9, the only SUMO E2-conjugating enzyme in *Drosophila*. As E3-ligases facilitate transfer of SUMO from the E2 enzyme to the final substrates they form complexes with E2 that are readily detected by co-immunoprecipitation (*Melchior et al., 2003*; *Pichler et al., 2002*; *Reverter and Lima, 2005*). Co-immunoprecipitation of tagged Bon and Ubc9 did not reveal an interaction (*Figure 6—figure supplement 1A*), suggesting that Bon does not act as an E3-ligase. Interestingly, K20R but not K9R Bon point mutant co-immunoprecipitates with Ubc9 (*Figure 6A*, *Figure 6—figure supplement 1B*), suggesting that inability to transfer SUMO from Ubc9 to Bon stabilizes the transiently formed complex between these proteins and the E3 SUMO-ligase potentially involved in the process.

Recently, Su(var)2–10 protein was identified as a Bon interactor in S2 cells (*Zhao et al., 2023*). We showed that Su(var)2–10 is an E3 SUMO-ligase that is required for suppressing tissue-inappropriate gene expression in the *Drosophila* germline, a function similar to the one we observed for Bon (*Ninova et al., 2020a*; *Ninova et al., 2020b*). To explore a possible interaction between Bon and Su(var)2–10, we first used co-immunoprecipitation, which confirmed binding of Bon and Su(var)2–10 (*Figure 6B*). Second, we tested Bon SUMOylation upon germline depletion of Su(var)2–10. Remarkably, knockdown of Su(var)2–10 led to a complete loss of Bon SUMOylation, indicating that Bon modification strongly depends on Su(var)2–10 (*Figure 6C*). Finally, germline depletion of Su(var)2–10 resulted in the derepression and loss of H3K9me3 signal on the reporter silenced by recruitment of Bon (*Figure 6D, E*). Combined, these results show that Bon modification and its repressive function depend on the E3 SUMO-ligase Su(var)2–10. Altogether, our results indicate that Su(var)2–10 promotes Bon SUMOylation and is required for Bon-induced H3K9me3 deposition and transcriptional silencing.

## Discussion

Bon, the only member of the TIF1 protein family in *Drosophila*, is ubiquitously expressed throughout development and previous studies have demonstrated its role in the development of several organs and tissues, particularly the nervous system (*Allton et al., 2009*; *Beckstead et al., 2001*; *Beckstead et al., 2005*; *Ito et al., 2012*; *Kimura et al., 2005*; *Salzberg et al., 1997*). We have found that

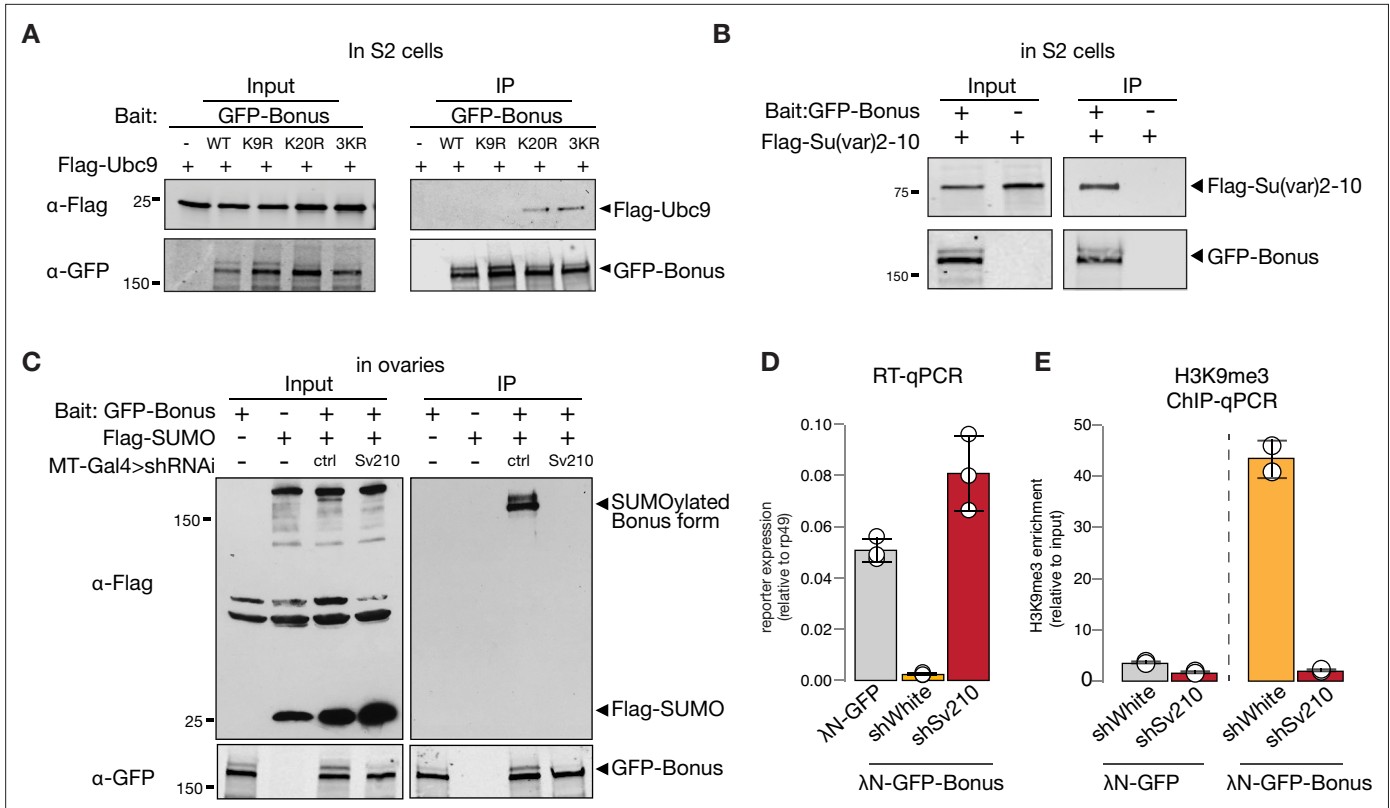

**Figure 6.** SUMO E3-ligase Su(var)2–10 interacts with Bonus and regulates its SUMOylation. (**A**) Western blot analysis shows the interaction between Bon and SUMO E2-conjugating enzyme Ubc9. Total protein lysates from S2 cells co-expressing Flag-Ubc9 and GFP-Bonus (WT), SUMO-deficient triple mutant 3KR or single mutated K9R, K20R were immunopurified using anti-GFP nanotrap beads. Lysate from cells not expressing GFP-tagged proteins was used as a negative control. (**B**) Bon interacts with Su(var)2–10. Western blot analysis of immunoprecipitation experiment using GFP nanotrap beads from S2 cells co-expressing GFP-Bonus and Flag-tagged Su(var)2–10. Lysate expressing only Flag-Su(var)2–10 was used as a negative control. (**C**) Western blot analysis shows the loss of SUMOylated Bon in fly ovaries upon Su(var)2–10 depletion. Total protein lysates from flies co-expressing *MT-Gal4*-driven Flag-SUMO and λ N-GFP-Bonus and shRNAs against Su(var)2–10 (Sv210) or control *white* (ctrl) were immunopurified using anti-GFP nanotrap beads. Ovarian lysates from flies expressing only Flag-SUMO, only expressing λ N-GFP-Bonus, or lacking Su(var)2–10 shRNA were used as controls. (**D**) Reporter repression by Bon depends on Su(var)2–10. Bar plot shows reporter expression (normalized to rp49 level) upon tethering of control λ N-GFP or λ N-GFP-Bonus in ovaries with Su(var)2–10 GLKD (shSv210), and control *white* GLKD (RT-qPCR, dots correspond to three independent biological replicates (n=3); error bars indicate st. dev.). (**E**) Bon H3K9me3 depositing requires Su(var)2–10. Bar plot shows H3K9me3 enrichment upon tethering of control λ N-GFP or λ N-GFP-Bonus in ovaries with Su(var)2–10 GLKD (shSv210), and control *white* GLKD (ChIP-qPCR, dots correspond to two independent biological replicates (n=2); error bars indicate st. dev.).

The online version of this article includes the following source data and figure supplement(s) for figure 6:

**Source data 1.** Annotated and uncropped western blots and raw images for *Figure 6A*.

**Source data 2.** Annotated and uncropped western blots and raw images for *Figure 6B*.

**Source data 3.** Annotated and uncropped western blots and raw images for *Figure 6C*.

**Figure supplement 1.** Bonus does not interact with Ubc9 in S2 cells.

**Figure supplement 1—source data 1.** Annotated and uncropped western blots and raw images for *Figure 6—figure supplement 1A*.

**Figure supplement 1—source data 2.** Annotated and uncropped western blots and raw images for *Figure 6—figure supplement 1B*.

depletion of Bon in the female germline leads to defective oogenesis resulting in rudimentary ovaries, loss of GSCs and sterility (*Figure 1D-F*). Germline knockdown of Bon resulted in the misexpression of hundreds of genes, which are normally restricted to other tissues, such as the nervous system and the gut (*Figure 2A,B*). Interestingly, Bon affects a diverse set of targets at different stages of oogenesis, as depletion of Bon using different germline drivers led to the ectopic expression of only partially overlapping sets of genes (*Figure 2—figure supplement 1C*). As Bon acts as a strong transcriptional repressor when recruited to a genomic locus (*Figure 3B*), genes upregulated upon Bon depletion might be its direct targets. However, the finding that only a fraction of these loci have

Bon-dependent H3K9me3 mark suggests that many of these genes are regulated by Bon indirectly, possibly through repression of other transcriptional regulators. Indirect regulation is further confirmed by the finding that germline depletion of Bon leads to enhanced expression of the *ple* gene in follicular cells surrounding the germline, where Bon levels were not perturbed (*Figure 2D*). Furthermore, it is possible that Bon-mediated H3K9me3 repression of its targets could play a prominent role during only early stages of oogenesis. To explore this possibility, it would be valuable to examine the changes of H3K9me3 mark upon Bon GLKD specifically during cyst formation and early nurse cell differentiation. Unfortunately, several attempts to map direct targets of Bon on chromatin using ChIP-seq turned out to be unsuccessful. Previous studies support the idea that Bon controls the expression of other developmental regulators that affect cell fate decisions. For example, a recent study has demonstrated that Bon together with the Hippo pathway is involved in cell fate decisions during eye development (*Zhao et al., 2023*). In the ovary, the Hippo pathway acts downstream of Hedgehog signaling to regulate follicle stem cells maintenance (*Hsu et al., 2017*). Bon has also been shown to control genes in the ecdysone response pathway (*Beckstead et al., 2001*), which in turn regulates multiple steps during oogenesis and controls the development of the *Drosophila* ovary (*Gancz et al., 2011*; *Hodin and Riddiford, 1998*; *König et al., 2011*).

The interesting unresolved question is how Bon identifies its genomic targets. In mammals, KAP-1 is recruited to chromatin through interaction with the large and diverse family of KRAB domain-containing C2H2-zinc-finger transcription factors (KRAB-ZFPs), which recognize their target DNA sequences – primarily various endogenous retroviruses and other types of TEs – via their zinc-finger domains (*Friedman et al., 1996*; *Margolin et al., 1994*; *Pengue et al., 1994*; *Vissing et al., 1995*; *Witzgall et al., 1994*). However, KRAB-ZFPs appeared during vertebrate evolution and are absent in insects. In agreement with this, we found that depletion of Bon did not activate the expression of TEs and instead affectes host genes (*Figure 2A*, *Figure 2—figure supplement 1A, B*). A different class of ZFPs, ZAD (Zinc-finger-associated domain)-zinc-finger proteins have expanded during insect evolution (*Chung et al., 2007*; *Chung et al., 2002*). It will be interesting to explore whether ZAD-ZFPs interact with and recruit Bon to its genomic targets, which if true, would represent a remarkable case of parallel evolution.

Our results reveal both similarities and differences between molecular mechanisms and functions of *Drosophila* Bon and mammalian TIF1 members, such as KAP-1. Similar to KAP-1, Bon induces transcriptional repression associated with accumulation of the H3K9me3 repressive histone mark (*Figure 3B, C*; *Khetchoumian et al., 2004*; *Nielsen et al., 1999*; *Schultz et al., 2002*; *Schultz et al., 2001*; *Venturini et al., 1999*). Furthermore, both proteins interact with the histone methyltransferase SetDB1 and a member of the NuRD histone deacetylase complex and these interactions are important for their repressive functions (*Figure 4*, *Figure 4—figure supplement 1A*; *Ito et al., 2012*; *Schultz et al., 2002*; *Schultz et al., 2001*; *Zhao et al., 2023*). Finally, the repressive function of both KAP-1 and Bon requires their post-translational modification by SUMO (*Figure 5G, H*; *Ivanov et al., 2007*; *Lee et al., 2007*; *Li et al., 2007*; *Mascle et al., 2007*). SUMOylation is a highly dynamic, reversible process, and SUMOylation of even a small fraction of a given protein was shown to drastically influence a protein's cellular function (reviewed in *Geiss-Friedlander and Melchior, 2007*; *Hay, 2005*). We found that SUMO-deficient Bon loses its association with chromatin and mislocalizes into discrete nuclear foci (*Figure 5E, F*). SUMOylation has been implicated in the assembly of functional nuclear condensates, such as PML bodies (*Ishov et al., 1999*; *Müller et al., 1998*; *Shen et al., 2006*; *Zhong et al., 2000*), yet Bon is more dispersed and associates with chromatin in its SUMOylated form and concentrates into foci in its unmodified form, suggesting that these foci might represent inactive Bon, which due to its lack of SUMOylation fails to form complexes with its partners such as SetDB1. An alternate possibility is that SUMOylation influences Bon's solubility, potentially preventing its aggregation. SUMOylation is known to enhance protein–protein interactions and promote efficient assembly of protein complexes during heterochromatin formation and transcriptional silencing (*Gareau and Lima, 2010*; *Gill, 2005*; *Gill, 2004*; *Jentsch and Psakhye, 2013*; *Martin et al., 2007*; *Shiio and Eisenman, 2003*).

Despite the universal role of SUMOylation in the repressive function of KAP-1 and Bon, there are crucial differences that suggests parallel and independent evolution in vertebrates and insects rather than conservation of an ancient mechanism. First, SUMOylation sites are not conserved between two groups (but conserved within each group) and in fact are located in different regions in Bon and

KAP-1. Bon is SUMOylated at a single residue, K20, close to its N terminus (*Figure 5C*) and this site seems to be conserved in other insects. In KAP-1 several SUMOylation sites are located in C-terminal bromodomain, the most prominent ones being K779 and K804 (*Ivanov et al., 2007*). Other mammalian TIF1 proteins are also SUMOylated at C-terminal region. SUMOylation of human TIF1α at lysine residues K723 and K741 was proposed to play a role in regulating genes involved in cell adhesion pathways (*Appikonda et al., 2018*), and multiple SUMO conjugations at lysines 776, 793, 796, and 839 of human TIF1γ were reported to be required for the transcriptional repression of TGFβ signaling (*Fattet et al., 2013*).

The second important difference is the molecular mechanism of SUMOylation of TIF1 members in insects and vertebrates. The PHD domain of mammalian KAP-1 functions as a SUMO E3-ligase to induce modification of the adjacent bromodomain (*Ivanov et al., 2007*). Importantly, evolutionary analysis of the TRIM/RBCC family suggests that the E3 SUMO-ligase activity has developed recently and only in a few specific members of this family and does not represent an ancient function of this protein family (*Marin, 2012*). In agreement with the evolutionary analysis, the lack of stable interaction between Bon and the E2-conjugating enzyme Ubc9 suggests that Bon does not have E3-ligase function (*Figure 6A*, *Figure 6—figure supplement 1A, B*). Instead, we found that Bon is SUMOylated by the distinct SUMO E3-ligase Su(var)2–10, which belong to the PIAS family of SUMO-ligases conserved among Metazoa (*Figure 6B, C*). Importantly, Su(var)2–10 is required for the repressive function of Bon. Thus, both the specific sites and the molecular mechanisms of SUMOylation of TIF1 proteins are different in insects and vertebrates, suggesting that TIF1 SUMOylation developed independently during evolution in these two groups.

In conclusion, our study reveals an essential function of *Drosophila* TIF1 factor, Bon, in repression of tissue-specific genes in the germline and suggests that Bon SUMOylation at a single site by the SUMO E3-ligase Su(var)2–10 is critical for its role as a transcriptional repressor.

## Materials and methods
### *Drosophila* fly stocks
All fly stocks and crosses were raised at 24°C. One- to two-day-old females were put on yeast for 1 day prior to dissection. Females from crosses with GLKD at early stages of oogenesis and respective control were 0–1 days old and were dissected right away. The following stocks were used: stocks with shRNAs targeting *Su(var)2–10* (shSv210, BDSC #32956), *Rpd3* (shRpd3, BDSC #33725), *Mi-2* (shMi-2, BDSC #35398) and *white* (shWhite, BDSC #33623) and *nos-Gal4;UAS-Cas9* (BDSC #54593) were obtained from the Bloomington *Drosophila* Stock Center. The fly line expressing sgRNAs targeting *bon* (VDRC #341851) was obtained from the Vienna *Drosophila* Resource Center. Fly lines UASp-mKate2-4xBoxB-K10polyA, UASp- $\lambda$ N-GFP-eGFP control, UASp- $\lambda$ N-GFP-Su(var)–10, shSmt3 were described previously (*Chen et al., 2016*; *Ninova et al., 2020a*). shSetDB1 was a gift from Julius Brennecke, the luciferase 8BoxB reporter and wild-type Oregon-R fly lines were a gift from Gregory Hannon, UASp-Flag-SUMO was a gift from Albert Courey. To obtain the shBonus fly lines, the short hairpin sequences (*Supplementary file 1*) was ligated into the pValium20 vector (*Ni et al., 2011*) and then integrated into the attP2 landing site (BDSC #8622). To generate the UASp- $\lambda$ N-GFP-Bonus and UASp- $\lambda$ N-GFP-Bonus[3KR] fly lines, full-length cDNA sequences of wild-type Bon or triple mutant Bon, respectively, were cloned in vectors containing a miniwhite marker followed by the UASp promoter sequence, and $\lambda$ N-GFP. Transgenic flies carrying these constructs were generated by phiC31 transformation by Best-Gene Inc and were integrated into the attP40 landing site (y$^1$ w$^{67}$c$^{23}$; P{CaryP}attP40). The expression of constructs was driven by maternal alpha-tubulin67C-Gal4 (*MT-Gal4*) (BDSC #7063), *nos-Gal4* (BDSC #4937), or *bam-Gal4* (BDSC #80579) drivers.

### Immunofluorescent microscopy and image processing
Seven to fifteen pairs of ovaries from *Drosophila* lines expressing UASP- $\lambda$ N-GFP-Bonus and UASP- $\lambda$ N-GFP-Bonus[3KR] under the control of the *MT-Gal4* driver were dissected in ice-cold phosphate-buffered saline (PBS) and then fixed in PBST solution (PBS, 0.1% Tween-20) supplemented with 4% formaldehyde for 20 min at room temperature with end-to-end rotation. Samples were washed three times 10 min with PBST and mounted in SlowFade Gold antifade Mountant with DAPI. Seven to fifteen pairs of ovaries from *Drosophila* Oregon-R flies and lines with GLKD of Bon under the control of

the *nos-Gal4* or *bam + nos* drivers were fixed in PBST supplemented with 4% formaldehyde for 20 min at room temperature with rotation and then washed three times 10 min with PBST. Fixed ovaries were incubated for 30 min with PBX (PBS, 0.1% Tween-20, 0.3% Triton X-100), and blocked in 5% normal goat serum (NGS) in PBX for 1 hr at room temperature. Samples were incubated with primary antibody in 3% NGS in PBX overnight at 4°C with rotation, followed by three washes in PBX solution for 10 min, and an overnight incubation with secondary antibody in 3% NGS in PBX at 4°C with rotation in the dark. After three washes in PBX, SlowFade Gold antifade Mountant with DAPI was added to the samples. Confocal images were acquired with a Zeiss LSM 800 using a ×63 oil immersion objective and were processed using Fiji. Primary antibodies to Vasa (rat, DSHB), to α-spectrin (mouse, 3A9 DSHB), and to Bon (a gift from Hugo Bellen) were used. Secondary antibodies were anti-mouse Alexa Fluor488, anti-rat Alexa Fluor546 (Invitrogen), and anti-guinea pig Cy3 (Jackson ImmunoResearch Inc).

## TUNEL assay
TUNEL analysis was performed using In Situ 'Cell Death Detection Kit' (TMR Red) (Roche, #12156792910).

## RNA in situ HCR
For RNA in situ HCR, probes, amplifiers, and buffers were purchased from Molecular Instruments (molecularinstruments.org) for *bon* (unique identifier: 4165/E324), *ple* (unique identifier: 4324/E516), *Rbp6* (unique identifier: 4408/E662), *pst* (unique identifier: 4408/E660), and *CG34353* (unique identifier: 4408/E658) transcripts. RNA in situ HCR v3.0 was performed according to the manufacturer's instructions for generic samples in solution.

## S2 cell line
*Drosophila* S2 cells (DGRC catalog #006) were cultured at 25°C in Schneider's *Drosophila* Medium supplemented with 10% heat-inactivated fetal bovine serum and 1× penicillin–streptomycin.

## Protein co-immunoprecipitation from S2 cells
S2 cells were transfected with plasmids encoding HA-, GFP-, and FLAG-tagged proteins under the control of the Actin promoter using TransIT-LT1 reagent (Mirus). 24–40 hr after transfection, cells were collected and resuspended in lysis buffer (20 mM Tris–HCl pH 7.4, 150 mM NaCl, 0.2% NP-40, 0.2% Triton X-100, 5% glycerol, 20 mM NEM (Sigma) and Complete Protease Inhibitor Cocktail (Roche)). The cell lysate was incubated on ice for 30 min, centrifuged and the supernatant collected. The supernatant was incubated with magnetic agarose GFP-Trap beads (Chromotek) for 2 hr at 4°C with end-to-end rotation. Beads were washed three to four times for 10 min with wash buffer (20 mM Tris–HCl pH 7.4, 0.1% NP-40, 150 mM NaCl) and boiled in 2× Laemmli buffer for 5 min at 95°C. The eluate was used for western blot analysis. For detection of SUMO-modified Bon, cells were lysed in RIPA (Radioimmunoprecipitation assay)-like buffer (20 mM Tris–HCl pH 7.4, 150 mM NaCl, 1% NP-40, 0.5% sodium deoxycholate, 0.1% SDS, 20 mM NEM, cOmplete Protease Inhibitor Cocktail), and washed in high salt wash buffer (20 mM Tris–HCl pH 7.4, 500 mM NaCl, 1% NP-40, 0.5% Sodium deoxycholate, 0.5% SDS, 20 mM NEM, cOmplete Protease Inhibitor Cocktail). For treatment with SUMO protease, cells were lysed in RIPA-like buffer without NEM and with in-house made SENP2 protease (20 mM Tris–HCl pH 7.4, 150 mM NaCl, 1% NP-40, 0.5% sodium deoxycholate, 0.1% SDS, 200 nM SENP2, cOmplete Protease Inhibitor Cocktail).

## Protein co-immunoprecipitation from fly ovaries
For detection of SUMO-modified Bon, 70–90 pairs of dissected ovaries co-expressing Flag-tagged SUMO and λ N-GFP-Bonus or λ N-GFP-Bonus[3KR] under the control of the *MT-Gal4* driver were lysed and dounced in 500 µl lysis buffer (20 mM Tris–HCl pH 7.4, 150 mM NaCl, 0.4% NP-40, 10% glycerol, 20 mM NEM, and cOmplete Protease Inhibitor Cocktail (Roche)). The ovary lysate was incubated on ice for 30 min, then centrifuged and the supernatant collected. The supernatant was incubated with magnetic agarose GFP-Trap beads (Chromotek) for 2–3 hr at 4°C with end-to-end rotation. Beads were washed four times ≥10 min at 4°C with ovary high salt wash buffer (20 mM Tris–HCl pH 7.4, 500 mM NaCl, 1% NP-40, 0.5% sodium deoxycholate, 0.5% SDS, 20 mM NEM, cOmplete Protease

Inhibitor Cocktail). Washed beads were further in 2× Laemmli buffer for 5 min at 95°C, and the eluate was analyzed by western blotting.

## Ovary fractionation

For whole ovaries and subcellular compartment extraction, 30–40 pairs of fly ovaries expressing $\lambda$ N-GFP-Bonus or $\lambda$ N-GFP-Bonus[3KR] were lysed and dounced in ice-cold 'AT' buffer (15 mM HEPES (4-(2-hydroxyethyl)-1-piperazineethanesulfonic acid)–NaOH pH 7.6, 10 mM NaCl, 5 mM MgOAc, 3 mM $CaCl_2$, 300 mM sucrose, 0.1% Triton X-100, 1 mM DTT (Dithiothreitol), cOmplete Protease Inhibitor Cocktail (Roche)). A small fraction of lysate was saved as whole cell lysate for western blot analysis. For subcellular fractionation 2 volumes of buffer 'B' (15 mM HEPES–NaOH pH 7.6, 10 mM NaCl, 5 mM MgOAc, 3 mM $CaCl_2$, 1 M sucrose, 1 mM DTT, cOmplete Protease Inhibitor Cocktail (Roche)) were added to the lysate. The lysate was incubated for 5 min on ice and then centrifuged at 5900 × $g$ for 15 min at 4°C. The supernatant was transferred to a new tube, centrifuged at 19,000 × $g$ for 10 min 4°C, and saved as cytoplasmic fraction for western blot analysis. The cell pellet was resuspended in 'E2' buffer (10 mM Tris–HCl pH 7.5, 200 mM NaCl, 1 mM EDTA (Ethylenediaminetetraacetic acid), 0.5 mM EGTA (ethylene glycol-bis(β-aminoethyl ether)-N,N,N',N'-tetraacetic acid), and cOmplete Protease Inhibitor Cocktail (Roche)) and centrifuged at 1500 × $g$ for 2 min at 4C. The supernatant was transferred to a new tube, centrifuged at 19,000 × $g$ for 10 min 4°C, and saved as nuclear fraction for western blot analysis. The pellet was once washed in E2 buffer, then resuspended in E2 buffer and incubated for 10 min at 4°C, followed by centrifugation at 1500 × $g$ for 2 min at 4°C. For chromatin extraction, the pellet was resuspended in 'E3' buffer (500 mM Tris–HCl pH 7.5, 500 mM NaCl, cOmplete Protease Inhibitor Cocktail (Roche)) and then sonicated for 5 min at high setting with 30sON/30sOFF in a Bioruptor sonicator (Diagenode) and centrifuged for 5 min at 19,000 × $g$ at 4°C. The supernatant was saved as chromatin fraction for western blot analysis. Protein concentration of all fractions was measured and further all saved fractions were boiled with 1× final concentration of Laemmli buffer for 5 min at 95°C and analyzed by western blotting.

## Western blotting

Proteins were separated by SDS–PAGE gel electrophoresis and transferred to a 0.45-µm nitrocellulose membrane (Bio-Rad) according to standard procedures. The membrane was blocked with 5% milk or with 0.2% I-block (Invitrogen) in PBST (PBS, 0.1% Tween-20) for 1 hr. The membrane was incubated with primary antibodies for 2 hr at room temperature or overnight at 4°C, followed by 3× washes for 5 min in PBST and incubation with secondary antibodies for 1 hr at room temperature. The membrane was washed three times for 5 min with PBST and then imaged with Odyssey system (Li-Cor). When primary antibody was HRP (horseradish peroxidase) conjugated, the membrane was washed 3 × 5 min with PBST, incubated with the HRP substrate, and X-ray film developed on an X-Ray Film Processor (Konica Minolta). The following antibodies were used: HRP-conjugated anti-FLAG (Sigma, A8592), mouse anti-FLAG (Sigma, F1804), rabbit polyclonal anti-GFP (*Chen et al., 2016*), rabbit anti-ubiquitin (abcam, ab134953), IRDye anti-rabbit and anti-mouse secondary antibodies (Li-Cor, #925–68070 and #925–32211).

conjugated, the membrane was washed 3 × 5 min with PBST, incubated with the HRP substrate, and X-ray film developed on an X-Ray Film Processor (Konica Minolta). The following antibodies were used: HRP-conjugated anti-FLAG (Sigma, A8592), mouse anti-FLAG (Sigma, F1804), rabbit polyclonal anti-GFP (*Chen et al., 2016*), rabbit anti-ubiquitin (abcam, ab134953), IRDye anti-rabbit and anti-mouse secondary antibodies (Li-Cor, #925–68070 and #925–32211).

## RNA extraction and RT-qPCR

For RNA extraction 10–20 pairs of dissected ovaries were homogenized in TRIzol (Invitrogen), RNA extracted, isopropanol precipitated, and treated with DNaseI (Invitrogen) according to the manufacturer's instructions. Reverse transcription was performed using random hexamer oligonucleotides with Superscript III Reverse Transcriptase (Invitrogen). qPCR was performed on a Mastercyclerep realplex PCR machine (Eppendorf). Three biological replicates per genotype were used for all RT-qPCR experiments. Target expression was normalized to rp49 mRNA expression. The data were visualized using Python 3 via JupyterLab. Primers used for qPCR analysis are listed in *Supplementary file 1*.

## RNA-seq and RNA-seq analysis

For RNA-seq libraries, total RNA was extracted from fly ovaries using TRIzol reagent. PolyA+ selection was performed using an NEBNext Poly(A) mRNA Magnetic Isolation Module (NEB, #E7490) for total RNA from lines with Bon GLKD driven by *nos-Gal4* (and matched siblings that lack the shRNA as control). Total RNA from line with Bon GLKD driven by *MT-Gal4* (and control flies that express an shRNA against the *white* gene) was depleted of ribosomal RNA with the Zymo-Seq RiboFree Total RNA Library Kit (Zymo Research, #R3000). RNA-seq libraries were made using the NEBNext Ultra II Directional RNA Library Prep kit for Illumina (NEB, #E7760) according to the manufacturer's instructions. Libraries were sequenced on the Illumina HiSeq 2500 platform. To quantify expression level of protein-coding genes and TEs, RNA-seq libraries were pseudoaligned to the *D. melanogaster* transcriptome (RefSeq, dm6) and transposon consensuses (from RepBase, *Jurka et al., 2005*), using kallisto (*Bray et al., 2016*). Differential expression analysis was done with sleuth using the gene analysis option (*Pimentel et al., 2017*). The average TPM (transcripts per million) between three biological replicas in knockdown versus control were calculated for fold changes in gene expression. For RNA-seq coverage tracks reads first were aligned to the *D. melanogaster* genome (dm6) using bowtie1 (v.1.2.2) allowing two mismatches and single mapping position. Tracks were generated using deep-Tools (v.3.5.1) bamCoverage function with 10 bp bin sizes.

GO biological-process term enrichment analysis was performed on the genes that were significantly derepressed upon Bon GLKD driven by *nos-Gal4* ($log_2FC > 1$, qval <0.05, LRT test, sleuth; *Pimentel et al., 2017*), using DAVID Bioinformatics Resources and all *Drosophila* genes that were not filtered out by sleuth as background. The enriched GO terms associated with two or less submitted genes were excluded. A significant threshold was applied using a multiple testing correction (Fisher's exact test p-value <0.01). The data visualization was performed using the 'ggplot2' R package.

## ChIP-qPCR and ChIP-seq

ChIP experiments were performed in two biological replicas as previously described (*Le Thomas et al., 2014*). In brief, 80–120 pairs of dissected ovaries were crosslinked with 1% formaldehyde in PBS for 10 min at room temperature, then quenched with Glycine (final concentration 25 mM). Frozen ovaries were dounced in RIPA buffer and then sonicated (Bioruptor sonicator) to a desired fragment sizes of 200–800 bp. Lysates were centrifuged at 19,000 × *g*, and supernatants collected. The supernatants were first precleared for 2 hr at 4°C using Protein G Dynabeads (Invitrogen). Precleared samples were immunoprecipitated with anti-H3K9me3 (abcam, ab8898) antibodies for 3–5 hr at 4°C, then 50 μl Protein G Dynabeads were added, and samples were further incubated overnight at 4°C. Beads were washed 3 × 10 min in LiCL buffer, followed by proteinase K treatment for 2 hr at 55°C and then overnight at 65°C. DNA was extracted by standard phenol–chloroform extraction. ChIP-qPCR was performed on the Mastercyclerep realplex PCR machine (Eppendorf). All ChIPs were normalized to respective inputs and to control region rp49. The data were visualized using Python 3 via JupyterLab. Primers used for qPCR analysis are listed in *Supplementary file 1*.

ChIP-seq libraries were prepared using NEBNext Ultra DNA Library Prep Kit Illumina (NEB) and sequenced on the Illumina HiSeq 2500 platform (PE 50 bp). After removal of the adaptors, reads with a minimal length of 18 nucleotides were aligned to the *D. melanogaster* genome (dm6) using bowtie1 (v.1.2.2) allowing two mismatches and single mapping position. Genome coverage tracks were generated using deepTools (v.3.5.1) bamCoverage function with 10 bp bin sizes. ChIP signal was normalized to input counts by calculating cpm (counts per million) using the deepTools bamCompare function with 50 bp bin sizes (ChIP/Input). Heatmaps were generated with deepTools plotHeatmap using normalized (ChIP/Input) BigWig files.

## Computational analysis

### Tissue specificity identification

To classify upregulated genes upon Bon GLKD driven by *nos-Gal4* ($log_2FC > 1$, qval <0.05, LRT test, sleuth; *Pimentel et al., 2017*), according to the tissues they are normally expressed in, we used RPKM values from the modENCODE anatomy RNA-seq dataset. The expression levels according to RPKM values from modENCODE anatomy RNA-seq dataset are no expression (0–0), very low (1–3), low (4–10), moderate (11–25), moderate high (26–50), high (51–100), very high (101–1000), and extremely high (>1000). The analysis of enrichment in each tissue was calculated as the number of

genes expressed at a certain expression level to the total number of provided genes. The data were visualized using Python 3 via JupyterLab.

## Phylogenetic analysis of Bon

The *D. melanogaster* Bon protein sequence was used to BLAST against the National Center for Biotechnology Information (NCBI) nonredundant protein database with the Position-Specific Iterated BLAST (PSI-BLAST) program. Orthologs of Bon in other *Drosophila* species and some insects were identified based on high sequence similarity. Multiple sequence alignment was performed using the ClustalW program. The species distribution of the orthologs was visualized using SeaView v5.0.5. Phylogenetic tree was built with ClustalW and iTOL v6 software.

## Acknowledgements

We thank members of the Aravin and Fejes Toth labs for discussion. We thank Peiwei Chen for suggesting some of the experiments. We appreciate the help of Anastasiya Grebin with the experiments. We are grateful to Julius Brennecke, Gregory Hannon, Albert Courey, the Bloomington Stock Center, and the Vienna *Drosophila* Resource Center for providing fly stocks, Hugo Bellen for providing antibodies. We thank Igor Antoshechkin (Millard and Muriel Jacobs Genetics and Genomics Laboratory, Caltech) for the help with sequencing, Giada Spigolon (Biological Imaging Facility, Caltech) for the help with microscopy, and Grace Shin for the help with HCR experiments. This work was supported by grants from the National Institutes of Health (R01 GM097363 to A.A.A. and R01 GM110217 to K.F.T. and R00 HD099316 to M.N.) and by the HHMI Faculty Scholar Award to A.A.A.

## Additional information

### Funding

| Funder | Grant reference number | Author |
| --- | --- | --- |
| National Institutes of Health | R01 GM097363 | Alexei Aravin |
| National Institutes of Health | R01 GM110217 | Katalin Fejes-Toth |
| Howard Hughes Medical Institute | Faculty Scholar Award | Alexei Aravin |
| National Institutes of Health | R00 HD099316 | Maria Ninova |

The funders had no role in study design, data collection, and interpretation, or the decision to submit the work for publication.

### Author contributions

Baira Godneeva, Conceptualization, Software, Formal analysis, Investigation, Visualization, Methodology, Writing – original draft; Maria Ninova, Conceptualization, Software, Formal analysis, Methodology, Writing – review and editing; Katalin Fejes-Toth, Alexei Aravin, Conceptualization, Formal analysis, Funding acquisition, Methodology, Writing – review and editing

### Author ORCIDs

Baira Godneeva ⓘ http://orcid.org/0009-0004-1662-8844
Alexei Aravin ⓘ http://orcid.org/0000-0002-6956-8257

Reviewer #1 (Public Review): https://doi.org/10.7554/eLife.89493.3.sa1
Reviewer #2 (Public Review): https://doi.org/10.7554/eLife.89493.3.sa2
Author Response https://doi.org/10.7554/eLife.89493.3.sa3

## Additional files

### Supplementary files
- MDAR checklist
- Supplementary file 1. List of primers.

### Data availability
The sequencing datasets have been deposited to the NCBI GEO archive under accession code GSE241375.

The following dataset was generated:

| Author(s) | Year | Dataset title | Dataset URL | Database and Identifier |
|---|---|---|---|---|
| Godneeva B | 2023 | SUMOylation of Bonus, the *Drosophila* homolog of Transcription Intermediary Factor 1, safeguards germline identity by recruiting repressive chromatin complexes to silence tissue-specific genes | https://www.ncbi.nlm.nih.gov/geo/query/acc.cgi?acc=GSE241375 | NCBI Gene Expression Omnibus, GSE241375 |

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
