## [Editor Report · eLife assessment]

This **important** study advances our knowledge of *Drosophila* Bonus, the sole ortholog of the mammalian transcriptional regulator Tif1. **Solid** evidence, both in vivo and in vitro, shows how SUMOylation controls the function of the Bonus protein and what the impact of SUMOylation on the function of Bonus protein in the ovary is.

---

## [Referee Report · Reviewer #1 (Public Review)]

Summary:

This important study from Godneeva et al. establishes a *Drosophila* model system for understanding how the activity of Tif1 proteins is modified by SUMO. The authors convincingly show that Bonus, like homologous mammalian Tif1 proteins, is a repressor, and that it interacts with other co-repressors Mi-2/NuRD and SetDB1 in Drosophia ovaries and S2 cells. They also show that Bonus is SUMOylated by Su(var)2-10 on one lysine at its N-terminus to promote its interaction with SetDB1. By combining biochemistry with an elegant reporter gene approach, they show that SUMOylation is important for Bonus interaction with SetDB1, and that this SUMO-dependent interaction triggers high levels of H3K9me3 deposition and gene silencing. While there are still major questions of how SUMO molecularly promotes this process, the authors conducted several experiments that will guide future work. For example, they showed that SUMOylation likely indirectly promotes Bon interaction with SetDB1 because mostly unSUMOylated Bon copurifies with SetDB1. They also show that SUMOylated and unSUMOylated Bon differentially localize within the cell, and preventing Bon SUMOylation alters its subcellular localization. These important experiments disfavor a simple model where SUMO bridges the Bon/SetDB1 interaction and hint at a more complex multi-step assembly process that regulates Bon-dependent transcriptional silencing.

---

## [Referee Report · Reviewer #2 (Public Review)]

Summary:

The authors analyze the functions and regulation of Bon, the sole *Drosophila* ortholog of the TIF1 family of mammalian transcriptional regulators. Bon has been implicated in several developmental programs, however the molecular details of its regulation have not been well understood. Here, the authors reveal the requirement of Bon in oogenesis, thus establishing a previously unknown biological function for this protein. Furthermore, careful molecular analysis convincingly established the role of Bon in transcriptional repression. This repressor function requires interactions with the NuRD complex and histone methyltransferase SetDB1, as well as sumoylation of Bon by the E3 SUMO ligase Su(var)2-10. Overall, this work represents a significant advance in our understanding of the functions and regulation of Bon and, more generally, the TIF1 family. Since Bon is the only TIF1 family member in Drosophila, the regulatory mechanisms delineated in this study may represent the prototypical and important modes of regulation of this protein family. The presented data are rigorous and convincing. As discussed below, this study can be strengthened by a demonstration of a direct association of Bon with its target genes, and by analysis of the biological consequences of the K20R mutation.

Strengths:

1. This study identified the requirement for Bon in oogenesis, a previously unknown function for this protein.

2. Identified Bon target genes that are normally repressed in the ovary, and showed that the repression mechanism involves the repressive histone modification mark H3K9me3 deposition on at least some targets.

3. Showed that Bon physically interacts with the components of the NuRD complex and SetDB1. These protein complexes are likely mediating Bon-dependent repression.

4. Identified Bon sumoylation site (K20) that is conserved in insects. This site is required for repression in a tethering transcriptional reporter assay, and SUMO itself is required for repression and interaction with SetDB1. Interestingly, the K20-mutant Bon is mislocalized in the nucleus in distinct puncta.

5. Showed that Su(var)2-10 is a SUMO E3 ligase for Bon and that Su(var)2-10 is required for Bon-mediated repression.

Weaknesses:

The study would be strengthened by demonstrating a direct recruitment of Bon to the target genes identified by RNA-seq. - It appears that the authors have attempted such an experiment, but it was not successful due to the current technical limitations, as the authors describe in their rebuttal.

The second area where the manuscript can be improved is to analyze the biological function of the K20R mutant Bonus protein. The molecular data suggest that this residue is important for function, and it would be important to confirm this in vivo. - Fig. 5G indeed shows that the 3KR mutant is deficient in inducing repression, which partially addresses this concern. In the future, it would be interesting to test if the single K20R is similarly deficient, and to analyze any resulting phenotypes.

---

## [Author Response]

The following is the authors’ response to the original reviews.

**Reviewer #1 (Public Review):**
Summary:This important study from Godneeva et al. establishes a *Drosophila* model system for understanding how the activity of Tif1 proteins is modified by SUMO. The authors nicely show that Bonus, like homologous mammalian Tif1 proteins, is a repressor, and that it interacts with other co-repressors Mi-2/NuRD and setdb1 in *Drosophila* ovaries and S2 cells. They also show that Bonus is SUMOylated by Su(var)2-10 on at least one lysine at its N-terminus to promote its interaction with setdb1. By combining nice biochemistry with an elegant reporter gene approach, they show that SUMOylation is important for Bonus interaction with setdb1, and that this SUMO-dependent interaction triggers high levels of H3K9me3 deposition and gene silencing. While there are still major questions of how SUMO molecularly promotes this process, this study is a valuable first step that opens the door for interesting future experimentation.Major Point:The RNAseq and ChIPseq data is not available. This is critical for the review of the paper and would help the readers and reviewers interpret the Bonus mutant phenotype and its mechanism of repressing genes.

The sequencing data have been deposited to the NCBI GEO archive. The accession number for all other RNA-seq and ChIP-seq data reported in this paper is GEO: GSE241375.

1. The author's conclusion that Bonus SUMOylation is "essential for its chromatin localization" is not supported by the data. Figure 5F shows less 3KR mutant in the chromatin fraction but there is still significant signal.

We appreciate the reviewer's feedback and agree that the term "essential" was not appropriate in this context. We have revised the manuscript to replace "essential" with "contributes to" to accurately reflect our findings.

1. The author's conclusion that Bonus is SUMOylated at a single site close to its N-terminus is not necessarily true. In several SUMO and Bonus blots throughout the paper (5B, 6C, S4A), there are >2 differentially migrating species that could represent more than one SUMO added to Bonus. While the single K20R mutation eliminates all of these species in Fig 5C, it is possible that K20R SUMOylation is required for additional SUMOylation events on other residues. One way to determine if Bonus is SUMOylated on multiple sites is to add recombinant SUMO protease to the extract and see if multiple higher molecular weight bands collapse into a single migrating species (implying multiple SUMOs) or multiple migrating species (implying something else is altering gel migration).

We appreciate the suggestion made by the reviewer. While we acknowledge the presence of occasional multiple bands in SUMO Western blots, the predominant pattern is the presence of unmodified Bon and a single additional band corresponding to SUMO-modified Bon. To investigate the possibility of multi-site SUMOylation, we performed requested experiment where we added SENP2 SUMO protease to the extract and checked Bon's SUMOylation. In the presence of NEM, we observed the unmodified form of Bon, as well as a single additional band representing a SUMO-modified form of Bon. Following SENP2 SUMO protease treatment, SUMOylation form of Bon was completely abolished in all samples, leaving only the unmodified Bon band (Extended Data Fig. 4D). This indicates that Bon is not SUMOylated on multiple sites and that the observed differential migration species likely result from other factors affecting gel migration.

1. The authors state that most upregulated genes in BonusGLKD are not highly enriched in H3K9me3. The heatmap in figure 3D is not an ideal presentation of this argument. The authors should show an example of what the signal on a highly enriched gene looks like for comparison. The authors also argue that because most upregulated genes in BonusGLKD are not highly enriched in H3K9me3, they must be indirectly repressed. Another possibility is that bonus-mediated H3K9me3 is only important (and present) during early nurse cell differentiation and is later lost and dispensable during the rapid endocycles. After bonus establishes repression though H3K9me3, it might be maintained through bonus-Mi2/Nurd, something else, or nothing at all. The authors could discuss this possibility or perform H3K9me3 ChIP during cyst formation and early nurse cell differentiation rather than in whole ovaries, which are enriched for later stages.

We thank the reviewer for their thoughtful comments and suggestions. In our revised manuscript we have included the tracks of gene that is highly enriched in H3K9me3 but remain unchanged upon Bon GLKD (Extended Data Fig. 3B). This addition allows for a visual comparison and better supports our argument that majority of genes upregulated in Bon GLKD are not enriched in H3K9me3 mark. We also appreciate the reviewer's suggestion regarding the potential temporal dynamics of Bon-mediated H3K9me3. It is indeed possible that Bon's role in establishing H3K9me3 might be more prominent during early nurse cell differentiation and less critical in later stages. We included discussion of this possibility in revised manuscript. To further explore it would be valuable to perform H3K9me3 ChIP during cyst formation and early nurse cell differentiation. However, given the limitations of our current resources and time limitations, we were unable to perform these experiments for the revised manuscript.

1. The BonusGLKD RNAseq analysis is underwhelming. The conclusion that "Bonus represses tissue-specific genes" has limited value. Every gene that is not expressed in ovaries is "tissue-specific." What subset of tissue-specific genes does Bonus repress? What common features do these genes have and how do they compare to other sets of tissue-specific genes, such as those reportedly repressed by setdb1, Polycomb proteins, small ovary, l(3)mbt, and stonewall (among others in female germ cells). Comparing these available data sets could help the authors understand the mechanism of Bonus repression and how BonusGLKD leads to sterility. The authors could also further analyze the differences between nos-Gal4 and MT-Gal4 to better understand why nos- but not MT-driven knockdown is sterile.

We appreciate the reviewer's feedback regarding the RNA-seq analysis and acknowledge the importance of identifying the specific subset of tissue-specific genes. The Figure 2C shows specifictissues where genes derepressed upon Bon GLKD are normally expressed. These are tissues/organs such as the head, digestive system, and nervous system. The reviewer's suggestion to compare our findings with existing datasets are valid and could indeed provide a more comprehensive understanding of Bon repression and its implications in female germ cells. However, many of the published datasets are based on mutant fly lines or use different GAL4 drivers to induce knockdowns, making direct comparisons challenging. We have conducted a preliminary analysis of available data, specifically nos-Gal4>SetDB1KD (GSE109852), and identified an overlap of 135 genes out of the 464 genes upregulated upon nos-Gal4>BonusKD with those affected by SetDB1 knockdown. We have included this result in the revised manuscript.

Main Study Limitations:1. It is unclear which genes are directly vs indirectly regulated by bonus, which makes it difficult to understand Bonus's repressive mechanism. Several lines of experiments could help resolve this issue. (1) Bonus ChIPseq, which the authors mentioned was difficult. (2) RNAseq of BonusGLKD rescued with KR3 mutation. This would help separate SUMO/setdb1-dependent regulation from Mi-2 dependent regulation. Similarly, comparing differentially expressed genes in Su(var)2-10GLKD, setdb1GLKD, 3KR rescue, and MI-2 GLKD could identify overlapping targets and help refine how bonus represses subsets of genes through these different corepressors.

We appreciate the reviewer's suggestions and agree that discrimination between direct and indirect Bon targets should be the next step in understanding Bon repressive mechanism. We have previously attempted to determine Bon direct targets using ChIP-seq approach. However, despite our multiple efforts using both native Bon antibodies and GFP-tagged Bon fly lines, analysis of ChIP-seq data did not reveal specific enrichment indicating that Bon – similar to many other chromatin-bound proteins – are not amenable to ChIP. The recommendation for RNA-seq analysis of Bon GLKD rescued with the 3KR mutation is valuable, and we will certainly consider it for future investigations.

We compared differentially expressed genes in Su(var)2-10 GLKD and Mi-2 GLKD and found limited overlap: out of the 231 genes affected by Bon GLKD, 39 genes were affected in Mi-2 GLKD and 42 in Su(var)2-10 GLKD. We acknowledge the importance of understanding which genes are directly or indirectly regulated by Bon and the potential for further experiments to address this question.

1. The paper falls short in discussing how SUMO might promote repression. This is important when considering the conservation (of lack thereof) of SUMOylation sites in Tif1 proteins in distantly related animals. One piece of data that was not discussed is the apparent localization of SUMOylated bonus in the cytoplasmic fraction of the blot in Figure 5F. Su(var)2-10 is mostly a nuclear protein, so is bonus SUMOylated in the nucleus and then exported to the cytoplasm? Also, setdb1 is a nuclear protein, so it is unlikely that the SUMOylated bonus directly interacts with setdb1 on target genes. Together with Fig 5E (unSUMOylatable Bonus aggregates in the nucleus), one could make a model where SUMO solubilizes bonus (perhaps by disassembling aggregates) and indirectly allows it to associate with setdb1 and chromatin. It is also important to note that in Figure 5I, the K3R mutation appears to lessen but not eliminate Bonus interaction with setdb1. This data again disfavors a model where SUMO establishes an interaction interface between setdb1 and Bonus. To determine which form of Bonus interacts with setdb1, the authors could perform a setdb1 pulldown and monitor the SUMOylation state of coIPed Bonus through mobility shift. If mostly unSUMOylated bonus interacts with setdb1, and SUMO indirectly promotes Bonus interaction with setdb1 (perhaps by disassembling Bonus aggregates), then the precise locations of Bonus SUMOylation sites could more easily shift during evolution, disfavoring the author's convergent evolution hypothesis.

We appreciate the reviewer's valuable feedback. Regarding the observation of SUMOylated Bon in the cytoplasmic fraction in Figure 5F, we recognize its significance. This finding has prompted us to consider a model in which SUMOylation may play a role in translocating Bon from the nucleus to the cytoplasm, potentially influencing interactions with SetDB1 and chromatin indirectly. Furthermore, Figure 5I which shows only a partial reduction in Bon-SetDB1 interaction with the 3KR mutation, suggests that SUMO may not be the primary mediator of this interaction. We recognize the need for further investigations to clarify SUMO's exact role in this context. In response to the reviewer's suggestion, we conducted SetDB1 pulldown experiments in S2 cells. The results reveal that indeed SetDB1 primarily interacts with unmodified Bon which is by far more abundant compared to SUMOylated form (Extended Data Fig. 5C). We think this experiment presents certain technical challenges, as the signal for Bon, when used as prey in co-IP experiments, is relatively faint, making it inherently difficult to detect the lower levels of SUMO-modified Bon. Additionally, in revised manuscript we have added new result of determining Bon interactors in ovary using mass-spec analysis, which showed that SetDB1 associates with wild-type, but not SUMO-deficient Bon. While our data support the idea that SUMO may contribute to Bon solubilization, possibly by disassembling aggregates, thereby indirectly facilitating its association with SetDB1 and chromatin, we acknowledge that the precise mechanism remains unclear.

**Reviewer #2 (Public Review):**
Summary:The authors analyze the functions and regulation of Bon, the sole *Drosophila* ortholog of the TIF1 family of mammalian transcriptional regulators. Bon has been implicated in several developmental programs; however, the molecular details of its regulation have not been well understood. Here, the authors reveal the requirement of Bon in oogenesis, thus establishing a previously unknown biological function for this protein. Furthermore, careful molecular analysis convincingly established the role of Bon in transcriptional repression. This repressor function requires interactions with the NuRD complex and histone methyltransferase SetDB1, as well as sumoylation of Bon by the E3 SUMO ligase Su(var)2-10. Overall, this work represents a significant advance in our understanding of the functions and regulation of Bon and, more generally, the TIF1 family. Since Bon is the only TIF1 family member in Drosophila, the regulatory mechanisms delineated in this study may represent the prototypical and important modes of regulation of this protein family. The presented data are rigorous and convincing. As discussed below, this study can be strengthened by a demonstration of a direct association of Bon with its target genes, and by analysis of the biological consequences of the K20R mutation.Strengths:1. This study identified the requirement for Bon in oogenesis, a previously unknown function for this protein.2. Identified Bon target genes that are normally repressed in the ovary, and showed that the repression mechanism involves the repressive histone modification mark H3K9me3 deposition on at least some targets.3. Showed that Bon physically interacts with the components of the NuRD complex and SetDB1. These protein complexes are likely mediating Bon-dependent repression.4. Identified Bon sumoylation site (K20) that is conserved in insects. This site is required for repression in a tethering transcriptional reporter assay, and SUMO itself is required for repression and interaction with SetDB1. Interestingly, the K20-mutant Bon is mislocalized in the nucleus in distinct puncta.5. Showed that Su(var)2-10 is a SUMO E3 ligase for Bon and that Su(var)2-10 is required for Bon-mediated repression.Weaknesses:The study would be strengthened by demonstrating a direct recruitment of Bon to the target genes identified by RNA-seq. Given that the global ChIP-seq was not successful, a few possibilities could be explored. First, Bon ChIP-qPCR could be performed on the individual targets that were functionally confirmed (e.g. rbp6, pst). Second, a global Bon ChIP-seq has been reported in PMID: 21430782 - these data could be used to see if Bon is associated with specific targets identified in this study. In addition, it would be interesting to see if there is any overlap with the repressed target genes identified in Bon overexpression conditions in PMID: 36868234.

We greatly appreciate the reviewer's suggestion to demonstrate the direct recruitment of Bon to the target genes. As described in our answer to reviewer #1, we attempted to determine Bon direct targets using ChIP-seq approach using both native Bon antibodies and GFP-tagged Bon fly lines. However, analysis of ChIP-seq data did not reveal specific enrichment. Similarly, Bon ChIP-qPCR on individual targets showed the same results suggesting that Bon – similar to many other chromatin-bound proteins – are not amenable to ChIP protocol, at least in standard conditions. To further explore this issue, we have analyzed results of a global Bon ChIP-seq reported in PMID: 21430782. We did not find Bon binding to individual targets, but even more importantly, we did not see clear Bon enrichment elsewhere in the genome confirming a conclusion that Bon targets on chromatin cannot be determined by ChIP. Additionally, we explored the possibility of overlap between target genes repressed by Bon in our study and those observed under Bon overexpression conditions in PMID: 36868234. While we did identify 41 genes in common, it's important to note that the datasets are derived from different tissues (pupal eyes vs. ovaries), making direct comparison problematic.

The second area where the manuscript can be improved is to analyze the biological function of the K20R mutant Bonus protein. The molecular data suggest that this residue is important for function, and it would be important to confirm this in vivo.

We appreciate the reviewer's suggestion to analyze the biological function of the K20R mutant Bon protein. While we acknowledge that we did not use single-site K20R mutant for in vivo experiments, we demonstrated that the mutant with the three-residue substitution (3KR) is incapable of inducing repression (Figure 5G). Given that other experiments consistently showed that K20 is the primarily SUMOylation site, this result supports the conclusion that K20 SUMOylation plays an important role in Bon-mediated transcriptional silencing.

**Reviewer #1 (Recommendations for The Authors):**
Make the RNAseq and ChIPseq data publicly available!

The sequencing data have been deposited to the NCBI GEO archive. The accession number for all other RNA-seq and ChIP-seq data reported in this paper is GEO: GSE241375.

**Reviewer #2 (Recommendations for The Authors):**
It would be interesting to identify the biological basis of aberrant ovary development in Bon depletion conditions. Previous studies (e.g. PMID: 11336699) suggested that Bon loss of function clones are cell lethal, and the developmental defects in oogenesis presented in the current study offer an opportunity to delve more into the causes of cell loss, e.g. by showing that the cells die via apoptosis.

Thank you for your valuable suggestion. In response to your comment, we performed a TUNEL assay to investigate whether germ cells in nos-Gal4>BonusKD ovaries undergo apoptosis. Our results indeed indicate that germ cells in these ovaries exhibit apoptosis, as evidenced by the TUNEL signal (Extended Data Fig. 1C). This information has been included in the revised manuscript to provide insights into the biological basis of aberrant ovary development in Bon depletion conditions.

The K20 residue could also be ubiquitinated. This possibility could at least be discussed, particularly given the presence of the RING Ub ligase domain in Bon that might potentially perform self-ubiquitination.

Indeed, the possibility that Bon can be ubiquitinated is a valid consideration. We have explored this possibility. We did not detect any signals with the Ubiquitin antibody in both wild-type Bon immunoprecipitant and triple-mutant [3KR] ovaries (in which K20 is also mutated) (Extended Data Fig. 4C). This suggests that K20 is more likely responsible for Bon SUMOylation rather than ubiquitination. We appreciate the reviewer's suggestion and have included this information into the revised manuscript.